# Association between obesity and risk of fracture, bone mineral density and bone quality in adults: A systematic review and meta-analysis

**Anne-Frédérique Turcotte**[1,2,3], **Sarah O'Connor**[4,5,6]**, Suzanne N. Morin**[7]**, Jenna C. Gibbs**[8]**, Bettina M. Willie**[9]**, Sonia Jean**[3,6]**, Claudia Gagnon**[1,2,3] *

**1** Endocrinology and Nephrology Unit, CHU de Québec-Université Laval Research Center, Québec (QC), Canada, **2** Obesity, Type 2 Diabetes and Metabolism Unit, Institut universitaire de cardiologie et de pneumologie de Québec–Université Laval Research Center, Québec (QC), Canada, **3** Department of Medicine, Faculty of Medicine, Laval University, Québec (QC), Canada, **4** Institut universitaire de cardiologie et de pneumologie de Québec–Université Laval Research Center, Québec (QC), Canada, **5** Department of Pharmacy, Faculty of Pharmacy, Laval University, Québec (QC), Canada, **6** Bureau d'information et études en santé des populations, Institut national de santé publique du Québec, Québec (QC), Canada, **7** Department of Medicine, Faculty of Medicine, McGill University, Montreal (QC), Canada, **8** Department of Kinesiology and Physical Education, McGill University, Montreal (QC), Canada, **9** Department of Pediatric Surgery, Shriners Hospital for Children-Canada, Research Centre, McGill University, Montreal (QC), Canada

* claudia.gagnon@crchudequebec.ulaval.ca

**Data Availability Statement:** All relevant data are within the paper and its Supporting Information files. The protocol of this systematic review and

## Abstract

### Background

The association between obesity and fracture risk may be skeletal site- and sex-specific but results among studies are inconsistent. Whilst several studies reported higher bone mineral density (BMD) in patients with obesity, altered bone quality could be a major determinant of bone fragility in this population.

### Objectives

This systematic review and meta-analysis aimed to compare, in men, premenopausal women and postmenopausal women with obesity vs. individuals without obesity: 1) the incidence of fractures overall and by site; 2) BMD; and 3) bone quality parameters (circulating bone turnover markers and bone microarchitecture and strength by advanced imaging techniques).

### Data sources

PubMed (MEDLINE), EMBASE, Cochrane Library and Web of Science were searched from inception of databases until the 13th of January 2021.

### Data synthesis

Each outcome was stratified by sex and menopausal status in women. The meta-analysis was performed using a random-effect model with inverse-variance method. The risks of hip

meta-analysis has been registered in the Prospective Register of Systematic Reviews (PROSPERO) (registration number: CRD42020159189).

**Funding:** No authors received specific funding for this work. AFT received a doctoral scholarship from the Fonds de recherche du Québec-Santé (FRQ-S) (2019-2020) and the Canadian Institutes for Health Research (CIHR) (2020-2023). SO received a doctoral scholarship from the FRQ-S (2019-2020) and CIHR (2020-2024). CG is a scholar from the FRQ-S and a recipient of a New investigator award from Diabetes Canada. JG is a recipient of a Natural Sciences and Engineering Research Council (NSERC) Discovery Grant and Early Career Researcher Launch Supplement. BW is supported by the Shriners Hospitals for Children and is a scholar from the FRQ-S. SNM is a scholar from the FRQ-S.

**Competing interests:** No authors have competing interests.

and wrist fracture were reduced by 25% (n = 8: RR = 0.75, 95% CI: 0.62, 0.91, P = 0.003, $I^2$ = 95%) and 15% (n = 2 studies: RR = 0.85, 95% CI: 0.81, 0.88), respectively, while ankle fracture risk was increased by 60% (n = 2 studies: RR = 1.60, 95% CI: 1.52, 1.68) in post-menopausal women with obesity compared with those without obesity. In men with obesity, hip fracture risk was decreased by 41% (n = 5 studies: RR = 0.59, 95% CI: 0.44, 0.79). Obesity was associated with increased BMD, better bone microarchitecture and strength, and generally lower or unchanged circulating bone resorption, formation and osteocyte markers. However, heterogeneity among studies was high for most outcomes, and overall quality of evidence was very low to low for all outcomes.

## Conclusions

This meta-analysis highlights areas for future research including the need for site-specific fracture studies, especially in men and premenopausal women, and studies comparing bone microarchitecture between individuals with and without obesity.

## Systematic review registration number

CRD42020159189

## Introduction

The incidence of fractures has been predicted to increase as the population is aging worldwide [1, 2]. Osteoporotic fractures are associated with excess mortality [3–5] in addition to being amongst the most frequent causes of disability and morbidity worldwide [6]. Consequently, fractures impose a financial burden on society in direct medical costs and indirect costs, which are projected to increase to $25.3 billion by 2025 in the United States [7]. Although the overall prevalence of fragility fractures is higher in women (especially in postmenopausal women) [8, 9], men generally have higher rates of fracture-related mortality [3].

Several clinical risk factors besides age, sex and menopausal status are known to affect fracture risk including a low body mass index (BMI) [10, 11]. Conversely, it still remains uncertain whether obesity is protective or not against fractures [12, 13]. Since obesity is projected to affect more than 50% of the population by 2030 [14, 15], it is imperative to determine how obesity should be considered in fracture risk assessment. The relationship between obesity and the risk of fracture is complex and appears to vary depending on skeletal site [16, 17], and may differ in men and women [11]. For example, a previous meta-analysis of the association of fracture risk and BMI in 398,610 women revealed that low BMI was a risk factor for hip and all osteoporotic fractures, but was a protective factor for lower leg fracture, whereas high BMI was a risk factor for humerus and elbow fractures [18].

Moreover, whilst numerous studies have consistently shown that areal bone mineral density (aBMD) is higher in patients with obesity [19], it appears that altered bone quality may be a major determinant of fracture risk in this population. Bone quality comprises bone microarchitecture, bone remodeling and bone tissue material properties, which includes bone strength, fracture toughness and fatigue strength. Bone strength can also be estimated through finite element analysis, which predicts bone resistance to stresses and strains. In recent years, few studies have evaluated the impact of obesity on bone microarchitecture and strength using advanced imaging techniques, such as peripheral quantitative computed tomography (pQCT)

and high resolution-pQCT (HR-pQCT) [20–22]. Some studies also reported lower bone turn-over in obesity, with a predominance of reduced bone formation over bone resorption [23, 24]. Besides, some studies also used obesity criteria other than BMI to assess the association between obesity and bone fragility [25–27]. Furthermore, coexistence of obesity with type 2 diabetes, which has also been associated with an increased risk of fracture [19], deteriorated bone microarchitecture (e.g., increased cortical porosity) and altered bone turnover [19, 28], may further impair bone health in individuals with obesity.

Previously published meta-analyses on the relationship between obesity and the risk of fractures targeted only women [18], hip fractures [29, 30], vertebral fractures [31], or overall fractures [32]. Moreover, no meta-analysis assessed whether bone quality parameters differ between adults with or without obesity. It is thus timely to summarize the available evidence and provide a more complete picture of bone health and fracture risk in men and women with obesity. The aims of this systematic review and meta-analysis were to compare, in men, pre-menopausal women and postmenopausal women with obesity vs. without obesity: 1) the inci-dence of fractures overall and by site; 2) BMD; and 3) bone quality parameters (i.e. bone microarchitecture and strength by advanced imaging techniques and circulating bone turn-over markers). Secondary aims were to investigate whether the presence of type 2 diabetes in people with obesity further affects fracture risk, BMD and bone quality parameters.

## Materials and methods

### Protocol and registration

We conducted this systematic review using the Cochrane review methodology [33], and reported our results according to the *Preferred reporting items for systematic review and meta-analysis (PRISMA) [34]*. The protocol was registered with the Prospective Register of System-atic Reviews (PROSPERO) on 28[th] April 2020 (registration number: CRD42020159189). Eligi-bility criteria and analysis were detailed and documented in the protocol. They are also described in the following sections of the manuscript.

### Eligibility criteria

Eligibility criteria were defined using an adaptation of the *PICOS* approach (Population, Expo-sure, Comparator, Outcomes and Study design) [34].

**Population.**  The study population were men and women of any ethnicity or setting. Only studies that included a majority of adults (i.e. at least 80% of the sample was aged 18 years or older, which is an arbitrary criterion commonly used in systematic reviews) [33] were selected, as findings among the paediatric population may be distinct due to ongoing bone development [35]. Studies including only individuals who experienced a fracture at baseline or had a joint replacement were excluded.

**Exposure.**  Studies were included when the exposure group was composed of individuals with obesity, characterized by an excessive fat accumulation that presents a risk to health. Any definition of obesity provided by the authors was considered. When multiple BMI categories were used, we used 25 kg/m$^2$ for threshold between obese/non-obese groups. Therefore, when results were reported for obese, overweight and normal-weight individuals, obese and over-weight individuals were combined in the "obesity" exposure group. Studies comparing equal categories (tertiles, quartiles or quintiles) were excluded since the ranges used were not comparable.

**Comparator.**  Studies were included when the comparison group was composed of indi-viduals without obesity. Any definition provided by the authors was considered.

**Outcomes.**   The primary outcomes were incident fractures at any or specific skeletal sites, that were either self-reported or confirmed by imaging. Secondary outcomes were: 1) aBMD at the total hip, femoral neck, lumbar spine and radius as well as volumetric BMD (vBMD) at the tibia and radius; 2) bone microarchitecture parameters [cortical thickness, cortical porosity, trabecular number, trabecular separation and trabecular connectivity, finite element modeling (FEM) estimated bone strength (failure load and stiffness) by pQCT or HR-pQCT]; and 3) circulating bone turnover markers [C-terminal telopeptide (CTX), N-terminal telopeptide (NTX), procollagen type 1 intact N-terminal propeptide (P1NP), osteocalcin and sclerostin]. Bone specific alkaline phosphatase, 25-hydroxyvitamin D and parathyroid hormone were not considered.

**Study design.**   For fracture outcomes, only studies using a prospective follow-up were considered; experimental studies with an intervention (e.g. nutrition, physical activity, bariatric surgery, pharmacotherapy, etc.) were excluded. For BMD, bone microarchitecture parameters and circulating bone turnover markers, all quantitative study designs, namely cross-sectional studies, cohort studies, clinical trials, case-control studies, retrospective studies, experimental studies and interrupted time series were considered. In longitudinal studies, only the baseline data were considered for secondary outcomes. Qualitative and descriptive studies, reviews, conference abstracts, letters to the editor or other non-peer reviewed publications were also excluded.

## Search strategy

Studies were identified by searching electronic databases, scanning the reference list of included studies and consulting experts in the field. The search was applied to PubMed (MEDLINE), EMBASE, Cochrane Library and Web of Science from inception of databases until the 1st of November 2019. The search was then updated on the 13th of January 2021 to ensure the most up-to-date review of the literature.

The search strategy (**S1 Table**) was revised by an information specialist (F. Bergeron) at Laval University, Québec City. Highly-sensitive and precision maximizing filters from the Evidence-Based Medicine (EBM) Toolkit form BMJ Best Practice were used for study design in PubMed and EMBASE [36]. No restriction was imposed on publication date, publication status or language. Results from the different databases were merged and duplicates were manually removed using EndNote X8.2 (Clarivate Analytics) reference software when the title, authors, journal and year of publication were identical.

## Study selection

Pilot testing was performed prior to the study selection process. Two reviewers (AFT and SO) independently screened titles and abstracts in duplicate to identify irrelevant manuscripts. Afterwards, eligibility assessment was performed independently by AFT and SO, in duplicate, using full-text reports. The eligibility process was conducted in an adapted electronic data collection form determined a priori and containing the inclusion and exclusion criteria described above. Multiple publications from the same studies were clustered. In case of uncertainty, AFT and SO deliberated to find consensus. In case of disagreement, a third reviewer (CG) was invited to the discussion. We assessed inter-reviewer agreement for full text selection using the kappa statistic. A kappa value of 0–0.20 was considered as no agreement, 0.21–0.39 was considered minimal agreement, 0.40–0.59 was considered weak agreement, 0.60–0.79 was considered moderate agreement, 0.80–0.90 was considered strong agreement, and 0.90 and above was considered perfect agreement [37]. The same selection process was used for the initial

search and the update. A flow diagram (**Fig 1**) from the PRISMA statement [34] was generated to map out the study selection process.

## Data extraction

A data collection form, adapted from the *Data collection form for RCTs* from Cochrane Airways and the *Cochrane Handbook of Systematic Reviews of Intervention* [38], was used. Pilot testing was performed on ten randomly-selected included studies, prior to the data extraction and amendments were made consequently. Data from the included studies were extracted independently in duplicate by AFT and SO. Disagreements were resolved by discussion between the two reviewers. CG was invited to the discussion if no agreement could be reached. In case of duplicate reporting, the reports with the largest number of participants were used. We tried to retrieve the missing data from the corresponding authors by sending emails twice. We contacted 8 authors for further information, among whom 3 authors responded.

A codification guide was generated to ensure the accuracy of the extraction process by the two reviewers. The following information was extracted from each included study: 1) study publication information (name of first author, year of publication, country of population); 2) population characteristics (total sample size, follow-up length (for fracture outcome only), size of exposure group, size of comparator group, mean age, sex, ethnicity, menopausal status, number of participants with type 2 diabetes, number of participants with a history of fracture, comorbidities or diseases affecting the participants and number of participants using medications known to affect bone metabolism); 3) exposure and comparator characteristics (group name, definition used); 4) outcomes characteristics (name of the outcome, reporting method for fractures (self-reported or confirmed), measurement tool and units of measurement); 5) measure of effect (type of effect, crude effect amplitude, crude 95% confidence interval and p-value, adjusted effect amplitude, covariates used in the adjusted model, adjusted 95% confidence interval and p-value). Two variables related to bone quality that provide information on bone strength, the estimated failure load and stiffness, were added after the beginning of the data extraction process. As those variables are estimated using finite element analysis, based on images captured by pQCT and HR-pQCT, we assumed they were already considered in the search strategy.

## Quality assessment

To verify the internal validity of included studies, AFT and SO independently assessed the risk of bias for each individual study. The Newcastle-Ottawa Scale (NOS) was used to evaluate the risk of bias for case-control and cohort studies [39]. The NOS tool assesses the quality of selection (4 items, 1 point each), comparability (1 item, 2 points) and outcome (3 items, 1 point each) of studies. The NOS tool generates a total score ranging from 0 (worst score) to 9 (best score). A score of 7 and above was considered low risk of bias, a score of 4–6 was considered moderate risk of bias and a score under 4 was considered high risk of bias [40]. The Joanna Briggs Institute (JBI) tool was used to assess the risk of bias for cross-sectional studies and for longitudinal studies from which we used cross-sectional data [41]. For each item, answers were either "Yes", "No", "Unclear" or "Not applicable". Scores ranged from 0 (worst score) to 8 (best score) and studies were judged as low risk of bias when the scores were above 6, moderate risk of bias when scores were between 4 and 6 and high risk of bias when scores were 3 or under [42]. Pilot testing was made on ten randomly-selected included studies to confirm adequate reliability prior to the risk of bias assessment, and amendments were made subsequently. Selection bias for each study was evaluated by verifying the eligibility criteria and selection of participants into the study. Confounding bias was assessed by evaluating if a confounding

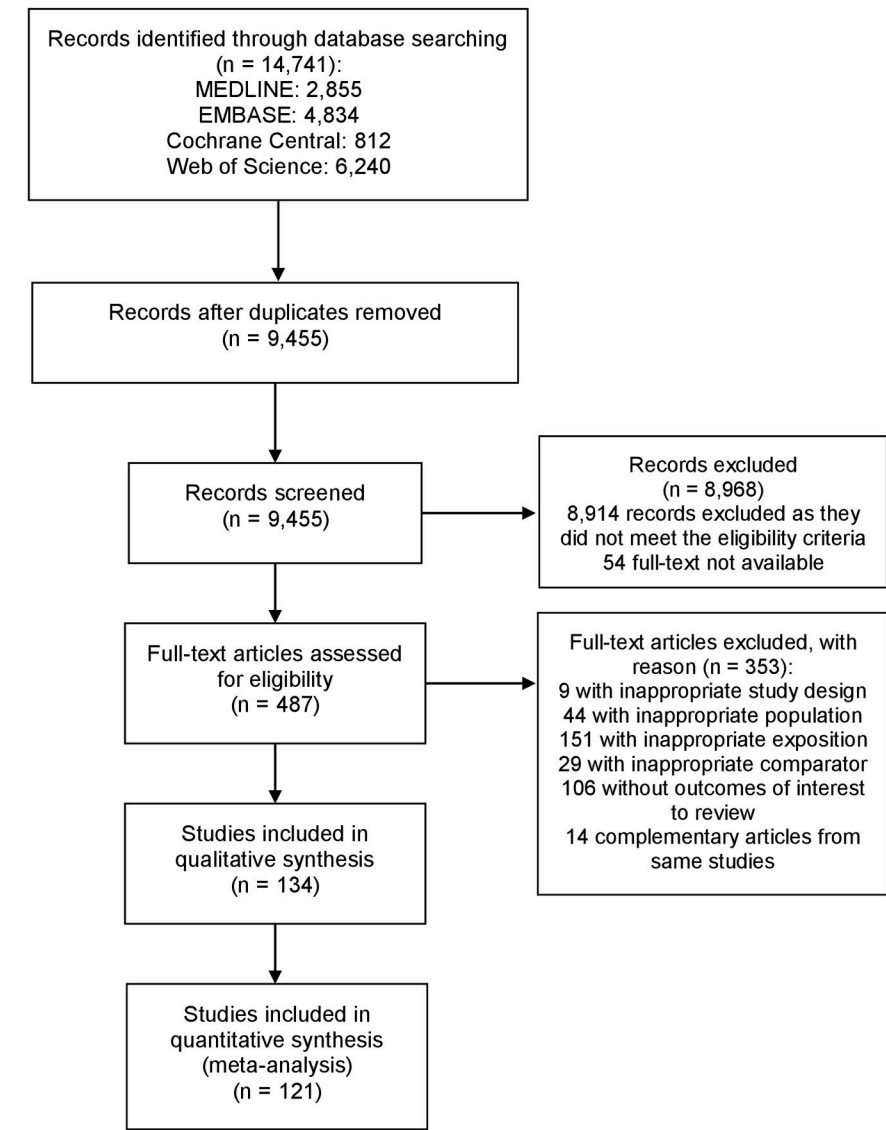

From: Moher D, Liberati A, Tetzlaff J, Altman DG, The PRISMA Group (2009). *Preferred Reporting Items for Systematic Reviews and Meta-Analyses*: The PRISMA Statement. PLoS Med 6(7): e1000097. doi:10.1371/journal.pmed1000097

**For more information, visit www.prisma-statement.org.**

**Fig 1.  Flow chart of study selection process.**

domain has not been measured at all or was not controlled for in the analysis. Information bias was evaluated by verifying if the exposure status was misclassified, if bias is introduced due to missing data, or if outcomes were misclassified or measured with error. Disagreements between AFT and SO were resolved by consensus.

## Statistical analyses

Descriptive analyses were completed to report characteristics of included studies, based on the *PICOS* approach. Moreover, descriptive synthesis was made for outcomes for which a meta-analysis could not be performed. Each outcome was evaluated comparing individuals with versus without obesity. When studies used different measures of effect size for an outcome, a transformation was performed whenever possible to enable comparison and combination of the studies for the meta-analysis. Relative risk (RR) with 95% confidence interval (CI) were used for fracture outcome. Mean differences (MD) with 95% CI were used to compare BMD at each bone site and bone quality parameters between the exposure and comparator groups. The meta-analysis was performed using a random-effect model with inverse-variance method, following the Cochrane review methodology for data analysis recommendations [43, 44]. For each outcome, estimates of the effect measure with their 95% CI are illustrated in forest plots.

All statistical analyses were performed with Review Manager software [45]. Each outcome was stratified based on sex and menopausal status (men, premenopausal women, postmenopausal women) since there are major differences in bone metabolism and risk of fracture between those populations [11, 31]. We included in the men or women's groups a mixed population when composed of at least 70% of either men or women. This arbitrary cut-off was chosen to minimise heterogeneity while maximizing statistical power within each group. When a mixed population included less than 70% of either men or women, men and women were combined and reported in a category called "studies combining men and women". In studies with multiple categories of obesity, we pooled groups together to allow comparison. We tested for heterogeneity with the $I^2$ statistic to measure inconsistency of the effects between studies [46]. $I^2$ over 50% was considered substantial heterogeneity and $I^2$ over 75% was considered considerable heterogeneity [47]. To explore potential causes of heterogeneity, subgroup analyses were planned a priori, and based on obesity cut-off criteria (as above, obesity criteria or overweight criteria according to the World health organization classification[44]), type 2 diabetes status (according to the author's definition), studies including or not individuals with comorbidities or use of medication known to affect bone metabolism, and overall risk of bias (low, moderate, high). After extraction of the data, only subgroup analyses based on obesity cut-off criterion and risk of bias were conducted since insufficient information on type 2 diabetes, presence of disease or use of medication was reported in these studies to allow analysis. A p-value <0.05 was considered statistically significant.

## Risk of bias across studies

Publication bias was assessed by visual evaluation of funnel plots [48, 49] produced by Review Manager software [45]. We evaluated the study mean differences for asymmetry, which can result from the non-publication of small studies with negative results. Quality of evidence for each outcome was assessed according to the Grading of Recommendations Assessment, Development and Evaluation (GRADE) approach [50]. The GRADE approach defines the quality of evidence based on within-study risk of bias (methodological quality), directness of evidence, heterogeneity, precision of effect estimates and risk of publication bias.

### Additional analyses

Sensitivity analyses were conducted to explore the robustness and accuracy of the results. To do so, studies were removed one at a time to explore how each study individually influenced the global estimate [51]. Sensitivity analysis based on the definition of exposure (BMI vs other obesity measures) could not be performed as the number of studies using measures other than BMI to define the exposure and comparator groups was too small.

## Results

### Study selection

The study selection process is described in **Fig 1**. We identified 14,741 citations through databases and screened 9,455 after the removal of duplicates. From those, 8,914 were discarded based on the title and abstract as they did not meet the eligibility criteria. Fifty-four studies were also discarded because the full-text was not available. The full-text of the remaining 487 reports was assessed for eligibility: 353 studies were excluded, leaving 134 for inclusion in the systematic review [20–22, 25, 52–181]. All included studies were in English or French. The kappa statistic was 0.82, displaying a strong inter-reviewer agreement for the full-text selection. Finally, 121 studies were included in the meta-analysis [20–22, 25, 52–153, 163–167, 169, 171–173, 175, 177–181]: 13 [154–162, 168, 170, 174, 176] were excluded because data was missing, could not be transformed, or could not be obtained from corresponding authors.

### Study characteristics

Study characteristics of the included studies are presented in **Table 1** (fracture), **Table 2** (aBMD and vBMD), **Table 3** (bone microarchitecture parameters) and **S2 Table** (circulating bone turnover markers). Moreover, **S3 Table** describes the methods used for measurement of bone turnover markers. All of the included studies were published between 1987 and 2021. Eighty-six studies selected for the review were cross-sectional studies, 34 were prospective cohort studies, 11 were case-control studies and 3 were epidemiological studies. Fifty-one studies were from Europe, 42 from Asia, 20 from North America, 8 from South America, 5 from Africa and 8 from Oceania. Fifty-six studies were conducted in postmenopausal women, 46 in a mixed population of men and women, 20 in premenopausal women and 12 in men. The studies included in this systematic review involved a total of 5,450,315 participants, including 2,798,344 individuals with obesity and 2,651,971 individuals without obesity. The mean age of the participants ranged between 18.2 and 78.3 years. Some information could not be retrieved from most of the studies such as the number of participants using medication or having comorbidities or diseases known to affect bone metabolism (e.g. diabetes), and the number of individuals with a history of fracture.

### Risk of bias within studies

The risk of bias assessment results for included studies are presented in **Tables 1–3** and **S2**. The overall risk of bias was considered "low" for 57 studies, "moderate" for 69 studies and "high" for 8 studies. The main criteria that were not reached for cross-sectional studies were: "the study subjects and setting described in detail" and "strategies to deal with confounding factors stated". In cohort studies, the quality criteria that received the lowest score were: "demonstration that outcome of interest was not present at start of study" and "was follow-up long enough for outcomes to occur".

**Table 1. Study and population characteristics of included studies for fracture outcome.**

| Study | Country | Study design (sample size) | Sample size by group | Obesity criterion | Inclusion obesity class II/ class III | Age (mean ± SD) | Sex (% female) | Incident fracture (N) | Follow up duration (years) | Fracture site(s) assessed | Fracture reporting method | Quality score[a] |
|---|---|---|---|---|---|---|---|---|---|---|---|---|
| **Postmenopausal women** | | | | | | | | | | | | |
| Armstrong 2012 | UK | Cohort (1,155,304) | OB: 619,621 / NO: 9,591 | OB: BMI≥25 / NO: 535,683 | No | OB: 56.1 ± 4.7 / NO: BMI<25 | 100 | OB: 11,168 / NO: 55.9 ± 4.8 | 8.3 | Overall, Hip, Wrist, Ankle | Adjudicated[c] | 7 |
| Compston 2011 | UK | Cohort (43,790) | OB: 10,441 / NO: 33,349 | OB: BMI≥30 / NO: 2,170 | No | OB: 67.0 ± 7.9 / NO: BMI<30 | 100 | OB: 633 / NO: 68.0 ± 8.6 | 2 | Overall, Hip, Clinical Vertebral, Wrist, Forearm, Ankle, Lower leg, Upper leg | Self-reported | 6 |
| Hermenegildo-Lopez 2021 | Spain | Cohort (1,185) | OB: 922 / NO: 263 | OB: BMI≥25 / NO: 17 | No | Overall 68.6 | 100 | OB: 37 / NO: BMI<25 | 2–4 | Overall | Self-reported | 6 |
| Kim 2017[b] | Korea | Cohort (2,625) | OB: 1,050 / NO: 1,575 | OB: %BF>33 / NO: 110 | No | OB: 56.7 ± 8.5 / NO: %BF<33 | 100 | OB: 93 / NO: 56.9 ± 8.9 | 9.4 | Overall | Self-reported | 6 |
| Kim 2018[b] | Korea | Cohort (138,288) | OB: 56,376 / NO: 81,912 | OB: BMI≥25 / NO: 1,442 | No | Overall 59.9 ± 7.4 | 100 | OB: 843 / NO: BMI<25 | 10.5 | Overall, Hip | Adjudicated | 8 |
| Luo 2020[b] | UK | Cohort (269,867) | OB: 164,195 / NO: 105,672 | OB: BMI≥25 / NO: BMI<25 | No | Range 40–69 | 100 | OB: 358 / NO: 267 | NR | Overall, Vertebral | Self-reported | 6 |
| Machado 2016 | Brazil | Cohort (433) | OB: 266 / NO: 19 | OB: BMI≥27 / NO: 167 | No | OB: 72.7 ± 5.7 / NO: BMI<27 | 100 | OB: 9 / NO: 74.9 ± 8.1 | 4.3 | Overall | Adjudicated | 7 |
| Meyer 2016[b] and Paik 2019[b] | USA | Cohort (41,677) | OB: 22,204 / NO: 63.1 | OB: WC≥88 / NO: 784 | No | OB: 64.7 / NO: 39,473 | 100 | OB: 404 / NO: WC<88 | 13 | Overall, Hip | Adjudicated | 8 |
| Rikkonen 2020 | UK | Cohort (12,715) | OB: 7,617 / NO: 5039 | OB: BMI≥25 / NO: 58.0 | No | OB: 58.0 / NO: 173 | 100 | OB: 249 / NO: BMI<25 | 18.3 | Overall, Hip | Adjudicated | 8 |
| Shen 2016[b] | Canada | Cohort (50,284) | OB: 30,702 / NO: 19,582 | OB: BMI≥25 / NO: 2,193 | Yes | OB: 66.0 ± 9.4 / NO: BMI 18.5–24.9 | 100 | OB: 2,341 / NO: 65.7 ± 10.2 | 6.2 | Overall, Hip | Adjudicated | 8 |

*(Continued)*

Table 1. (Continued)

| Study | Country | Study design (sample size) | Sample size by group | Obesity criterion | Inclusion obesity class II/ class III | Age (mean ± SD) | Sex (% female) | Incident fracture (N) | Follow up duration (years) | Fracture site(s) assessed | Fracture reporting method | Quality score[a] |
|---|---|---|---|---|---|---|---|---|---|---|---|---|
| Sogaard 2016[b] | Norway | Cohort (29,240) | OB: 18,987 / NO: 10,253 | OB: BMI≥25 / NO: 64.6 | No | OB: 65.3 / NO: 715 | 100 | OB: 888 / NO: BMI<25 | 8.4 | Overall, Hip | Adjudicated | 8 |
| Tanaka 2013 | Japan | Cohort (1,479) | OB: 348 / NO: 1,131 | OB: BMI≥25 / NO: 879 | No | OB: 63.2 ± 10.1 / NO: BMI 18.5–24.9 | 100 | OB: 337 / NO: 62.5 ± 11.2 | 6.7 | Overall, Hip, Clinical Vertebral, Forearm, Humerus | Adjudicated | 6 |
| **Premenopausal women** | | | | | | | | | | | | |
| Huopio 2005 | Finland | Cohort (3,078) | OB: 839 / NO: 2,239 | OB: BMI≥28 / NO: 202 | No | Range 47–56 | 100 | OB: 72 / NO: BMI<28 | 3.6 | Overall | Adjudicated and self-reported | 7 |
| Jordan 2013[b] | Thailand | Cohort (25,401) | OB: 3,238 / NO: 613 | OB: BMI≥25 / NO: 22,163 | No | Range 19–49 | 100 | OB: 124 / NO: BMI<25 | 4 | Overall | Self-reported | 7 |
| **Men** | | | | | | | | | | | | |
| Jordan 2013[b] | Thailand | Cohort (24,024) | OB: 5,974 / NO: 18,050 | OB: BMI≥25 / NO: 849 | No | Range 19–49 | 0 | OB: 248 / NO: BMI<25 | 4 | Overall | Self-reported | 7 |
| Kim 2017[b] | Korea | Cohort (2,189) | OB: 876 / NO: 1,313 | OB: % BF>22 / NO: 50 | No | OB: 56.4 ± 8.6 / NO: 54.9 ± 8.8 | 0 | OB: 27 / NO: % BF<22 | 9.4 | Overall | Self-reported | 6 |
| Kim 2018[b] | Korea | Cohort (142,070) | OB: 48,958 / NO: 93,112 | OB: BMI≥25 / NO: 1,069 | No | Overall 59.9 ± 7.4 | 0 | OB: 277 / NO: BMI<25 | 10.5 | Overall, Hip | Adjudicated | 8 |
| Luo 2020[b] | UK | Cohort (226,945) | OB: 170,192 / NO: BMI<25 | OB: BMI≥25 / NO: 119 | No | Range 40–69 | 0 | OB: 351 / NO: 56,753 | NR | Overall, Vertebral | Self-reported | 6 |
| Meyer 2016[b] and Paik 2019[b] | USA | Cohort (35,488) | OB: 12,421 / NO: 23,067 | OB: WC≥101 / NO: 65.6 | No | OB: 67.3 / NO: 314 | 0 | OB: 169 / NO: WC101 | 13 | Overall, Hip | Adjudicated | 8 |
| Nielson 2011 | USA | Cohort (5,918) | OB: 4,290 / NO:1,628 | OB: BMI≥25 / NO: 325 | Yes | OB: 72.9 ± 5.3 / NO: 75.0 ± 6.4 | 0 | OB: 710 / NO: BMI<25 | 7 | Overall, Hip, Upper limb, Lower limb | Adjudicated | 8 |

(Continued)

**Table 1.** (Continued)

| Study | Country | Study design (sample size) | Sample size by group | Obesity criterion | Inclusion obesity class II/ class III | Age (mean ± SD) | Sex (% female) | Incident fracture (N) | Follow up duration (years) | Fracture site(s) assessed | Fracture reporting method | Quality score[a] |
|---|---|---|---|---|---|---|---|---|---|---|---|---|
| Scott 2017 | Australia | Epidemiological (1,486) | OB: 631 / NO: 855 | OB: %BF≥30 / NO: 87 | No | OB: 78.0 ± 6.5 / NO: %BF<30 | 0 | OB: 66 / NO: 78.3 ± 7.8 | 5 | Overall | Adjudicated | 6 |
| Shen 2016[b] | Canada | Cohort (4,627) | OB: 3,177 / NO: 1,450 | OB: BMI≥25 / NO: 146 | Yes | OB: 68.1 ± 9.8 / NO: BMI 18.5–24.9 | 0 | OB: 195 / NO: 69.9 ± 10.8 | 4.7 | Overall, Hip | Adjudicated | 8 |
| Sogaard 2016[b] | Norway | Cohort (32,109) | OB: 22,236 / NO: 9,873 | OB: BMI≥25 / NO: 66.4 | No | OB: 65.1 / NO: 413 | 0 | OB: 538 / NO: BMI<25 | 8.4 | Overall, Hip | Adjudicated | 8 |
| **Mixed population** | | | | | | | | | | | | |
| Huang 2018 | China | Cohort (21,262) | OB: 10,404 / NO: 10,858 | OB: BMI≥24 / NO: 169 | No | 40+ | 48.4 | OB: 118 / NO: BMI<24 | 8 | Overall, Hip | Adjudicated | 7 |
| Prieto-Alhambra 2012 | Spain | Cohort (1,111,352) | OB: 843,997 / NO: 265 | OB: BMI≥25 / NO: 267,355 | Yes | NR | 52.1 | OB: 695 / NO: BMI<25 | 3 | Overall | Adjudicated | 7 |
| Rousseau 2016 | Canada | Retrospective (177,464) | OB: 50,704 / NO: 3,375 | NR | Yes | OB: 42.7 ± 11 / NO: 126,760 | 72.3 | OB: 1,145 / NO: 42.6 ± 11 | 4.4 | Overall, Hip, Upper limb, Distal lower limb | Adjudicated | 6 |
| Scott 2016 | Australia | Cohort (2,134) | OB: 781 / NO: 1,353 | OB: BMI≥25 / NO: BMI<25 | No | OB: 62.3 ± 6.8 / NO: 268 | 50.9 | OB: 146 / NO: 62.4 ± 7.5 | 5–10 | Overall | Self-reported | 6 |
| Kouvonen 2013 | Finland | Cohort (69,515) | OB: 30,678 / NO: 38,837 | OB: BMI≥25 / NO: BMI<25 | No | Range 17–67 | 80 | NR | 7.8 | Overall | Adjudicated | 7 |
| Wolinsky 2009 | USA | Cohort (5,291) | OB: 2,756 / NO: 2,535 | OB: BMI≥25 / NO: BMI<25 | No | 69+ | 62 | NR | NR | Hip | Adjudicated | 5 |

OB: obese; NO: non-obese; BMI: Body-mass Index; WC: Waist circumference; %BF: percentage body fat; NR: Not reported.

BMI is expressed in kg/m$^2$.

WC is expressed in cm.

[a]Quality score was obtained from the Newcastle-Ottawa Scale (NOS) (<4: high risk of bias; 4–6 moderate risk of bias; ≥7 low risk of bias).

[b]These studies fall into two subgroup categories (postmenopausal women, premenopausal women, men) as results were stratified by sex.

[c]Fractures confirmed through database linkage, radiography or other methods.

**Table 2. Study and population characteristics of included studies for bone mineral density outcome.**

| Study | Country | Study design (sample size) | Sample size by group | Obesity criterion | Age (mean ± SD) | Sex (% female) | BMD assessment tool | Site of BMD assessment | Quality score[a] |
|---|---|---|---|---|---|---|---|---|---|
| **Postmenopausal women** | | | | | | | | | |
| Al-Shoumer 2012 | Kuwait | CS (454) | OB: 403 | OB: BMI≥25 | Range 50–89 | 100 | DXA | Total Hip, Femoral Neck, Lumbar Spine | 5 |
| | | | NO: 51 | NO: BMI<25 | | | | | |
| Asli 2020 | Iran | CS (260) | OB: 177 | OB: BMI≥25 | OB: 61.5 ± 9.1 | 89.6 | DXA | Total Hip, Femoral Neck, Lumbar Spine, Radius | 6 |
| | | | NO: 83 | NO: BMI<25 | NO: 61.4 ± 8.9 | | | | |
| Bilic-Curcic 2017 | Croatia | CS (114) | OB: 83 | OB: BMI>27 | ≥45 | 100 | DXA | Femoral Neck, Lumbar Spine | 5 |
| | | | NO: 31 | NO: BMI≤27 | | | | | |
| Chain 2021[b] | Brazil | CS (255) | OB: 154 | OB: Body fat≥40% | OB: 53.8 ± 8.2 | 100 | DXA | Femoral Neck, Lumbar Spine | 4 |
| | | | NO: 101 | NO: Body fat<40% | NO: 52.1 ± 7.8 | | | | |
| Dytfeld 2011 | Poland | CS (92) | OB: 66 | OB: WC≥80 | 69.5 ± 7.3 | 100 | DXA | Femoral Neck, Lumbar Spine | 5 |
| | | | NO: 26 | NO: WC<80 | | | | | |
| Glogowska-Szelag 2019 | Poland | CS (80) | OB: 40 | OB: BMI 30–34.9 | NR | 100 | DXA | Lumbar Spine | 4 |
| | | | NO: 40 | NO: BMI 18–24.9 | | | | | |
| Holecki 2007 | Poland | Case-control (62) | OB: 43 | NR | OB: 50.1 ± 4.5 | 100 | DXA | Lumbar Spine | 6 |
| | | | NO: 19 | | NO: 53.8 ± 5.2 | | | | |
| Ibrahim 2011 | Egypt | CS (74) | OB: 37 | OB: BMI>30 | OB: 57.4 ± 4.4 | 100 | DXA | Femoral Neck, Lumbar Spine | 7 |
| | | | NO: 37 | NO: BMI<25 | NO: 56.6 ± 3.5 | | | | |
| Jiajue 2014 | China | CS (1,410) | OB: 810 | OB: BMI≥25 | OB: 64.0 ± 15.3 | 100 | DXA | Femoral Neck, Lumbar Spine | 5 |
| | | | NO: 600 | NO: BMI<25 | NO: 65.6 ± 15.9 | | | | |
| Khukhlina 2019 | Ukraine | CS (60) | OB: 30 | NR | OB: 63.9 ± 1.2 | 70 | DXA | Total Hip, Femoral Neck | 4 |
| | | | NO: 30 | | NO: 56.5 ± 3.0 | | | | |
| Kim 2016 | Korea | CS (124) | OB: 52 | OB: BMI≥25 | OB: 60.2 ± 6.7 | 100 | DXA | Total Hip, Femoral Neck, Lumbar Spine | 8 |
| | | | NO: 72 | NO: BMI<25 | NO: 59.6 ± 7.4 | | | | |
| Korpelainen 2003 | Finland | CS (1,222) | OB: 815 | OB: BMI≥28.5 | OB: 72.1 ± 1.2 | 100 | DXA | Radius | 7 |
| | | | NO: 407 | NO: BMI<28.5 | NO: 72.1 ± 1.7 | | | | |
| Machado 2016 | Brazil | Cohort (433) | OB: 266 | OB: BMI>27 | OB: 72.7 ± 5.7 | 100 | DXA | Total Hip, Femoral Neck, Lumbar Spine | 7 |
| | | | NO: 167 | NO: BMI<27 | NO: 74.9 ± 8.1 | | | | |
| Mazocco 2017 | Brazil | CS (392) | OB: 299 | OB: BMI≥25 | 59.6 ± 8.2 | 100 | DXA | Total Hip, Femoral Neck, Lumbar Spine | 6 |
| | | | NO: 93 | NO: BMI 18.5–24.9 | | | | | |
| Mendez 2013 | Mexico | CS (813) | OB: 690 | OB: BMI≥25 | OB: 59.6 ± 14.0 | 100 | DXA | Total Hip, Femoral Neck, Lumbar Spine | 7 |
| | | | NO: 123 | NO: BMI<25 | NO: 59.6 ± 7.5 | | | | |
| Messina 2019 | Italy | CS (60) | OB: 30 | OB: WC>88 | OB: 68 ± 10 | 100 | DXA | Lumbar Spine | 6 |
| | | | NO: 30 | NO: WC≤88 | NO: 63 ± 9 | | | | |
| Olmos 2018 | Spain | Cohort (2,597) | OB: 2094 | OB: BMI≥25 | OB: 65.4 ± 13.4 | 70.3 | DXA | Total Hip, Femoral Neck, Lumbar Spine | 6 |
| | | | NO: 503 | NO: BMI<25 | NO: 61.0 ± 10.2 | | | | |
| Papakitsou 2004 | Greece | CS (130) | OB: 104 | OB: BMI≥25 | 55.5 (range: 54.2–56.7) | 100 | DXA | Femoral Neck, Lumbar Spine | 7 |
| | | | NO: 26 | NO: BMI<25 | | | | | |

*(Continued)*

**Table 2.** (Continued)

| Study | Country | Study design (sample size) | Sample size by group | Obesity criterion | Age (mean ± SD) | Sex (% female) | BMD assessment tool | Site of BMD assessment | Quality score[a] |
|---|---|---|---|---|---|---|---|---|---|
| Povoroznyuk 2017 | Ukraine | CS (566) | OB: 230 | OB: BMI≥30 | OB: 64.5 ± 8.2 | 100 | DXA | Femoral Neck, Lumbar Spine, Radius | 6 |
| | | | NO: 336 | NO: BMI<30 | NO: 64.2 ± 8.1 | | | | |
| Ribot 1987 | France | CS (176) | OB: 77 | NR | OB: 53.2 ± 6.0 | 100 | DXA | Lumbar Spine | 1 |
| | | | NO: 99 | | NO: 53.1 ± 5.7 | | | | |
| Scott 2020[b] | Australia | Cohort (1,692) | OB: 1424 | OB: BMI≥30 | OB: 70.0 ± 0.1 | 100 | DXA | Total Hip | 7 |
| | | | NO:268 | NO: BMI<30 | NO: 70.0 ± 0.1 | | | | |
| Shaarawy 2003 | Egypt | CS (90) | OB: 37 | OB: BMI>30 | 58.8 ± 0.5 | 100 | DXA | Lumbar Spine | 4 |
| | | | NO: 53 | NO: BMI 20–25 | | | | | |
| Shiraki 1991 | Japan | CS (65) | OB: 22 | OB: BMI≥25 | OB: 72.8 ± 8.0 | 100 | DXA | Radius | 5 |
| | | | NO: 43 | NO: BMI 20–24.9 | NO: 75.3 ± 5.9 | | | | |
| Shayganfar 2020 | Iran | CS (1361) | OB: 1134 | OB: BMI≥25 | 56.4 ± 10.4 | 77.6 | DXA | Femoral Neck, Lumbar Spine | 5 |
| | | | NO: 337 | NO: BMI<25 | | | | | |
| Silva 2007 | Brazil | Retrospective CS (588) | OB: 299 | OB: BMI≥25 | OB: 54.5 ± 3.7 | 100 | DXA | Femoral Neck, Lumbar Spine | 4 |
| | | | NO: 289 | NO: BMI<25 | NO: 53.9 ± 4 | | | | |
| Sornay-Rendu 2013 | France | Case-control (189) | OB: 63 | OB: BMI≥30 | OB: 68.6 ± 7 | 100 | DXA, HR-pQCT | Total Hip, Lumbar Spine, Radius, Tibia | 8 |
| | | | NO: 126 | NO: BMI 18.5–24.9 | NO: 68.2 ± 7.4 | | | | |
| Tajik 2013 | Malaysia | CS (297) | OB: 218 | OB: BMI≥25 | OB: 56.2 ± 6.5 | 100 | DXA | Femoral Neck, Lumbar Spine | 7 |
| | | | NO: 79 | NO: BMI<25 | NO: 56.1 ± 4.1 | | | | |
| Tanaka 2013 | Japan | Cohort (1,479) | OB: 348 | OB: BMI≥25 | OB: 63.2 ± 10.1 | 100 | DXA | Femoral Neck, Lumbar Spine | 5 |
| | | | NO: 1131 | NO: BMI 18.5–24.9 | NO: 62.5 ± 11.2 | | | | |
| Tarquini 1997 | Italy | CS (95) | OB: 60 | OB: BMI≥25 | OB: 59.5 ± 6.3 | 100 | DXA | Radius | 5 |
| | | | NO: 35 | NO: BMI<25 | NO: 58.3 ± 8.8 | | | | |
| Tay 2018 | USA | Cohort (30) | OB: 10 | OB: BMI≥30 | OB: 65.3 ± 9.3 | 70 | DXA | Total Hip, Femoral Neck, Lumbar Spine, Radius | 7 |
| | | | NO: 20 | NO: BMI<30 | NO: 61.7 ± 13.4 | | | | |
| Wu 2016 | China | CS (212) | OB: 88 | OB: BMI>25 | OB: 64.4 ± 5.3 | 100 | DXA | Femoral Neck, Lumbar Spine | 4 |
| | | | NO: 124 | NO: BMI<25 | NO: 63.5 ± 4.7 | | | | |
| Zhou 2010 | China | CS (1,479) | OB: 750 | OB: BMI≥25 | OB: 57.5 ± 7.4 | 100 | DXA | Total Hip, Femoral Neck, Lumbar Spine | 5 |
| | | | NO: 729 | NO: BMI<25 | NO: 56.8 ± 5.8 | | | | |
| **Premenopausal women** | | | | | | | | | |
| Baheiraei 2005 | Australia | CS (88) | OB: 65 | OB: BMI≥25 | 48.5 ± 8.3 | 100 | DXA | Femoral Neck, Lumbar Spine | 5 |
| | | | NO: 23 | NO: BMI<25 | | | | | |
| Bachmann 2014 and Schorr 2019 | USA | CS (122) | OB: 53 | OB: BMI≥25 | OB: 26.5 ± 5.6 | 100 | DXA | Total Hip, Femoral Neck, Lumbar Spine, Radius | 7 |
| | | | NO: 69 | NO: BMI 18.5–24.9 | NO: 26.7 ± 6.2 | | | | |
| DeSimone 1990 | USA | CS (216) | OB: 51 | OB: >30% ideal body weight | OB: 67.0 ± 14.3 | 100 | DXA | Femoral Neck, Lumbar Spine, Radius | 2 |
| | | | NO: 67.5 ± 16.3 | NO: 165 | NO: ≤30% ideal body weight | | | | |
| El Hage 2014 | Lebanon | CS (3,989) | OB: 2708 | OB: BMI≥25 | OB: 62.3 ± 11.8 | 100 | DXA | Radius | 3 |
| | | | NO: 1281 | NO: BMI<25 | NO: 56.8 ± 12.6 | | | | |

(*Continued*)

**Table 2.** (*Continued*)

| Study | Country | Study design (sample size) | Sample size by group | Obesity criterion | Age (mean ± SD) | Sex (% female) | BMD assessment tool | Site of BMD assessment | Quality score[a] |
|---|---|---|---|---|---|---|---|---|---|
| Gafane 2015 | South Africa | Epidemiological (434) | OB: 261 | OB: BMI≥25 | OB: 61.6 ± 8.6 | 100 | DXA | Radius | 8 |
| | | | NO: 173 | NO: BMI<25 | NO: 59.5 ± 7.1 | | | | |
| Indhavivadhana 2015 | Thailand | CS (427) | OB: 208 | OB: WC≥80 | 52.6 ± 5.4 | 100 | DXA | Femoral Neck, Lumbar Spine | 5 |
| | | | NO: 219 | NO: WC<80 | | | | | |
| Jang 2016 | Korea | CS (1,296) | OB: 263 | OB: BMI≥23 | 32.8 ± 3.9 | 100 | DXA | Total Hip, Lumbar Spine | 5 |
| | | | NO: 1033 | NO: BMI<23 | | | | | |
| Kumar 2016 | India | CS (234) | OB: 95 | OB: BMI≥23 | NR | 100 | DXA | Femoral Neck, Lumbar Spine | 5 |
| | | | NO: 139 | NO: BMI<23 | | | | | |
| Liel 1988 | USA | CS (182) | OB: 42 | OB: >30% ideal body weight | OB: 37.0 ± 10.2 | 100 | DXA | Femoral Neck, Lumbar Spine, Radius | 2 |
| | | | NO: 140 | NO: 34.5 ± 11.8 | NO: ≤30% ideal body weight | | | | |
| Lim 2019 | Korea | CS (143) | OB: 54 | OB: BMI≥25 | OB: 21.4 ± 1.0 | 100 | DXA | Femoral Neck, Lumbar Spine | 8 |
| | | | NO: 89 | NO: BMI<25 | NO: 21.0 ± 1.2 | | | | |
| Liu 2014 | USA | CS (471) | OB: 281 | OB: BMI≥25 | OB: 48.6 ± 17.8 | 100 | DXA | Total Hip, Femoral Neck, Lumbar Spine, Radius | 6 |
| | | | NO: 190 | NO: BMI<25 | NO: 35.8 ± 11.8 | | | | |
| Maimoun 2020 | France | CS (152) | OB: 38 | OB: BMI≥30 | OB: 21.3 ± 2.9 | 100 | DXA | Total Hip, Lumbar Spine, Radius | 7 |
| | | | NO: 38 | NO: BMI<30 | NO: 21.0 ± 3.2 | | | | |
| Maimoun 2020 | France | CS (318) | OB: 139 | OB: BMI≥30 | OB: 47.0 ± 15.2 | 100 | DXA | Total Hip, Lumbar Spine, Radius | 7 |
| | | | NO: 40 | NO: BMI<30 | NO: 45.6 ± 16.9 | | | | |
| Pereira 2007 | Brazil | CS (27) | OB: 16 | OB: BMI≥30 | OB: 37.8 ± 1.7 | 100 | DXA | Femoral Neck, Lumbar Spine, Radius | 6 |
| | | | NO: 11 | NO: BMI<30 | NO: 37.2 ± 3.1 | | | | |
| Pollock 2011 | USA | CS (48) | OB: 15 | OB: Body fat≥32% | OB: 19.0 ± 1.1 | 100 | pQCT | Radius, Tibia | 6 |
| | | | NO: 33 | NO: Body fat<32% | NO: 19.3 ± 1.3 | | | | |
| Pollock 2007 | USA | CS (115) | OB: 22 | OB: Body fat≥32% | OB: 18.4 ± 0.5 | 100 | pQCT | Radius, Tibia | 8 |
| | | | NO: 93 | NO: Body fat<32% | NO: 18.2 ± 0.4 | | | | |
| Segall-Gutierrez 2013 | USA | CS (15) | OB: 10 | OB: BMI≥30 | 20–35 | 100 | DXA | Lumbar Spine | 6 |
| | | | NO: 5 | NO: BMI 18.5–24.9 | | | | | |
| Sukumar 2011 | USA | Case-control (111) | OB: 52 | OB: BMI>35 | OB: 52.7 ± 11.7 | 100 | DXA | Total Hip, Femoral Neck, Lumbar Spine | 8 |
| | | | NO: 59 | NO: BMI<27 | NO: 50.6 ± 8.5 | | | | |
| Takata 1999 | Japan | CS (51) | OB: 20 | OB: BMI>25 | OB: 52.8 ± 13.4 | 100 | DXA | Total Hip, Lumbar Spine | 3 |
| | | | NO: 31 | NO: BMI 21–25 | NO: 54.7 ± 15.4 | | | | |
| Wampler 2005 | USA | CS (1,568) | OB: 970 | OB: BMI≥25 | Range 50–79 | 100 | DXA | Total Hip, Femoral Neck, Lumbar Spine | 5 |
| | | | NO: 598 | NO: BMI<25 | | | | | |
| Wang 2020[b] | China | CS (1,272) | OB: 502 | OB: BMI≥25 | OB: 50.4 ± 12.1 | 100 | DXA | Radius | 6 |
| | | | NO: 770 | NO: BMI<25 | NO: 44.8 ± 14.3 | | | | |
| Wiacek 2010 | Poland | CS (4,359) | OB: 2984 | OB: BMI≥25 | Range 40–79 | 100 | DXA | Femoral Neck | 3 |
| | | | NO: 1375 | NO: BMI<25 | | | | | |

(*Continued*)

**Table 2.** (Continued)

| Study | Country | Study design (sample size) | Sample size by group | Obesity criterion | Age (mean ± SD) | Sex (% female) | BMD assessment tool | Site of BMD assessment | Quality score[a] |
|---|---|---|---|---|---|---|---|---|---|
| Zantut-Wittmann | Brazil | Cohort (52) | OB: 22 | OB: BMI≥25 | Range 20–39 | 100 | DXA | Total Hip, Femoral Neck, Lumbar Spine | 6 |
|  |  |  | NO: 30 | NO: BMI<25 |  |  |  |  |  |
| **Men** |  |  |  |  |  |  |  |  |  |
| Ayoub 2017 | Lebanon | CS (67) | OB: 44 | OB: BMI≥25 | OB: 22.4 ± 3.6 | 0 | DXA | Total Hip, Femoral Neck, Lumbar Spine | 7 |
|  |  |  | NO: 23 | NO: BMI 18.5–24.9 | NO: 22.2 ± 2.8 |  |  |  |  |
| Chain 2021[b] | Brazil | CS (249) | OB: 136 | OB: Body fat≥30% | OB: 51.7 ± 7.9 | 0 | DXA | Femoral Neck, Lumbar Spine | 4 |
|  |  |  | NO: 113 | NO: Body fat<30% | NO: 54.2 ± 7.9 |  |  |  |  |
| Choi 2015 | Korea | CS (1,089) | OB: 368 | OB: BMI≥25 | 58.8 ± 7.5 | 0 | DXA | Total Hip, Femoral Neck | 7 |
|  |  |  | NO: 721 | NO: BMI<25 |  |  |  |  |  |
| Jiang 2015 | China | CS (358) | OB: 219 | OB: BMI≥24 | 72.8 ± 9.5 | 0 | DXA | Total Hip, Femoral Neck, Lumbar Spine | 5 |
|  |  |  | NO: 139 | NO: BMI<24 |  |  |  |  |  |
| Kanazawa 2008 | Japan | CS (163) | OB: 73 | OB: BMI≥24 | OB: 56.8 ± 21.0 | 0 | DXA | Femoral Neck, Lumbar Spine, Radius | 7 |
|  |  |  | NO: 90 | NO: BMI<24 | NO: 58.6 ± 15.3 |  |  |  |  |
| Kang 2014 | China | CS (502) | OB: 365 | OB: BMI≥24 | OB: 61.3 ± 23.6 | 0 | DXA | Total Hip, Femoral Neck, Lumbar Spine | 7 |
|  |  |  | NO: 137 | NO: BMI<24 | NO: 64.7 ± 17.1 |  |  |  |  |
| Nielson 2011 and Shen 2015 | USA | CS (3,067) | OB: 2238 | OB: BMI≥30 | OB: 72.8 ± 7.8 | 0 | DXA | Total Hip | 8 |
|  |  |  | NO: 829 | NO: BMI<30 | NO: 74.5 ± 6.3 |  |  |  |  |
| Salamat 2013 | Iran | CS (230) | OB: 135 | OB: BMI≥25 | OB: 61.7 ± 8.1 | 0 | DXA | Total Hip, Femoral Neck, Lumbar Spine | 6 |
|  |  |  | NO: 95 | NO: BMI<25 | NO: 63.9 ± 7.9 |  |  |  |  |
| Scott 2017 | Australia | Epidemiological (1,486) | OB: 631 | OB: body fat≥30% | OB: 78.0 ± 6.5 | 0 | DXA | Total Hip | 6 |
|  |  |  | NO: 855 | NO: Body fat<30% | NO: 78.3 ± 7.8 |  |  |  |  |
| Scott 2020[b] | Australia | Cohort (1,719) | OB: 1503 | OB: BMI≥30 | OB: 70.0 ± 0.1 | 0 | DXA | Total Hip | 7 |
|  |  |  | NO:216 | NO: BMI<30 | NO: 70.0 ± 0.1 |  |  |  |  |
| Tencerova 2019 | Denmark | CS (54) | OB: 35 | OB: BMI≥25 | OB: 34.8 ± 2.6 | 0 | DXA | Total Hip, Femoral Neck, Lumbar Spine | 7 |
|  |  |  | NO: 19 | NO: BMI<25 | NO: 31.0 ± 3.0 |  |  |  |  |
| Wang 2020[b] | China | CS (850) | OB:472 | OB: BMI≥25 | OB: 45.5 ± 14.1 | 0 | DXA | Radius | 5 |
|  |  |  | NO: 378 | NO: BMI<25 | NO: 45.8 ± 16.2 |  |  |  |  |
| **Mixed population** |  |  |  |  |  |  |  |  |  |
| Amarendra Reddy 2009 | India | CS (303) | OB: 151 | OB: BMI>25 | OB: 28.0 ± 7.7 | 50.8 | DXA | Total Hip, Femoral Neck, Lumbar Spine, Radius | 6 |
|  |  |  | NO: 152 | NO: BMI≤25 | NO: 27.7 ± 8.8 |  |  |  |  |
| Andersen 2014 | Denmark | CS (72) | OB: 36 | OB: BMI>30 | OB: 41± 8 | 66.7 | DXA, HR-pQCT | Lumbar Spine, Radius, Tibia | 7 |
|  |  |  | NO: 36 | NO: BMI 19.5–24.8 | NO: 40.1 ± 7.8 |  |  |  |  |
| Buta 2012 | Romania | CS (67) | OB: 43 | OB: BMI≥25 | OB: 48.7 ± 16.8 | 100 | DXA | Lumbar Spine | 6 |
|  |  |  | NO: 24 | NO: BMI<25 | NO: 47.8 ± 9.4 |  |  |  |  |
| De Araujo 2017 | Brazil | Case-control (78) | OB: 54 | NR | OB: 53.0 ± 13.6 | 57.7 | DXA | Total Hip, Femoral Neck, Lumbar Spine | 3 |
|  |  |  | NO: 24 |  | NO: 55.0 ± 7.0 |  |  |  |  |

(*Continued*)

**Table 2.** (Continued)

| Study | Country | Study design (sample size) | Sample size by group | Obesity criterion | Age (mean ± SD) | Sex (% female) | BMD assessment tool | Site of BMD assessment | Quality score[a] |
|---|---|---|---|---|---|---|---|---|---|
| Dubois 2003 | Netherlands | CS (28) | OB: 14 | OB: BMI≥25 | OB: 60 ± 14.9 | 50 | DXA | Total Hip, Femoral Neck, Lumbar Spine | 6 |
| | | | NO: 14 | NO: BMI<25 | NO: 61 ± 14.4 | | | | |
| Evans 2015 | UK | CS (223) | OB: 146 | OB: BMI≥30 | OB: 49.8 ± 9.9 | 50.7 | DXA, HR-pQCT | Lumbar Spine, Radius, Tibia | 8 |
| | | | NO: 77 | NO: BMI 18.5–24.9 | NO: 49.8 ± 9.8 | | | | |
| Gandham 2020 | Australia | Cohort (1,099) | OB: 303 | OB: BMI≥30 | OB: 62.5 ± 7.2 | 51.2 | DXA | Total Hip, Lumbar Spine | 6 |
| | | | NO: 796 | NO: BMI<30 | NO: 62.2 ± 7.6 | | | | |
| Kao 1994 | China | CS (343) | OB: 158 | OB: BMI>25 | NR | 72.3 | DXA | Lumbar Spine | 5 |
| | | | NO: 185 | NO: BMI<25 | | | | | |
| Kin 1991 | Japan | CS (812) | OB: 163 | OB: BMI≥25 | 20+ | 77.5 | DXA | Lumbar Spine | 6 |
| | | | NO: 649 | NO: BMI<25 | | | | | |
| Kirchengast 2002 | Austria | CS (119) | OB: 64 | OB: BMI≥25 | 71.7 ± 7.7 | 56.3 | DXA | Femoral Neck | 6 |
| | | | NO: 55 | NO: BMI<25 | | | | | |
| Lim 2013 | Korea | Cohort (25) | OB: 16 | OB: BMI≥25 | OB: 23.3 ± 0.2 | 52 | DXA | Total Hip, Femoral Neck, Lumbar Spine | 8 |
| | | | NO: 9 | NO: BMI≤25 | NO: 24.6 ± 0.3 | | | | |
| Lloyd 2016 | USA | CS (2,570) | OB: 1718 | OB: BMI≥25 | OB: 73.4 ± 4.0 | 50.8 | DXA | Total Hip, Femoral Neck | 7 |
| | | | NO: 852 | NO: BMI<25 | NO: 73.9 ± 2.9 | | | | |
| Rudman 2019 | UK | CS (342) | OB: 243 | OB: BMI≥25 | 62.5 ± 0.5 | 55.6 | DXA | Femoral Neck, Lumbar Spine | 4 |
| | | | NO: 99 | NO: BMI 18.5–24.9 | | | | | |
| Scott 2016 | Australia | Cohort (2,134) | OB: 781 | NR | OB: 63.6 ± 10.2 | 50.8 | DXA | Total Hip, Lumbar Spine | 5 |
| | | | NO: 1353 | | NO: 63.3 ± 11.0 | | | | |
| Scott 2018 | Australia | CS (168) | OB: 79 | OB: BMI≥30 | 67.8 ± 12.0 | 53.8 | pQCT | Tibia | 6 |
| | | | NO: 89 | NO: BMI<30 | | | | | |
| **Studies not included in the meta-analysis** | | | | | | | | | |
| Bener 2005 | Qatar | CS (649) | OB: 303 | OB: BMI≥30 | NR | 100 | DXA | Femoral Neck, Lumbar Spine | 8 |
| | | | NO: 346 | NO: BMI<30 | | | | | |
| Dickey 2006 | Ireland | CS (328) | OB: 143 | OB: BMI≥25 | OB: 46 | 60.1 | DXA | Femoral Neck, Lumbar Spine | 3 |
| | | | NO: 185 | NO: BMI 20–24.9 | NO: 48 | | | | |
| Gojkovic 2020 | Serbia | CS (1974) | OB: 1395 | OB: BMI≥25 | Range 54–76 | 94.5 | DXA | Femoral Neck, Lumbar Spine | 5 |
| | | | NO: 579 | NO: BMI<25 | | | | | |
| Gomez-Cabello 2013 | Spain | CS (223) | NR | OB: BMI≥25 | Rage 65–89 | 71.3 | DXA | Femoral Neck, Lumbar Spine | 8 |
| | | | | NO: BMI<25 | | | | | |
| Jawhar 2020 | Malaysia | Cohort (635) | NR | OB: BMI≥25 | 60.0 ± 11.5 | 100 | DXA | Total Hip, Femoral Neck | 4 |
| | | | | NO: BMI<25 | | | | | |
| Vandevyver 1997 | Belgium | CS (748) | OB: 190 | OB: BMI≥30 | 70.8 | NR | DXA | Femoral Neck | 3 |
| | | | NO: 558 | NO: BMI<30 | | | | | |
| Yoon 2019 | Korea | CS (2552) | OB: 1510 | OB: BMI≥23 | ≥50 | 0 | DXA | Femoral Neck | 5 |
| | | | NO: 1042 | NO: BMI<23 | | | | | |

CS: cross-sectional; OB: obese; NO: non-obese; BMI: Body Mass Index; WC: Waist circumference.

BMI is expressed in kg/m$^2$.

WC is expressed in cm.

[a]Quality score was obtained from the Joanna Briggs Institute tool (JBI): <4: high risk of bias; 4–6 moderate risk of bias; ≥7 low risk of bias.

[b]These studies fall into two subgroup categories (postmenopausal women, premenopausal women, men) as results were stratified by sex.

**Table 3. Study and population characteristics of included studies for bone microarchitecture outcome by peripheral quantitative computed tomography (pQCT) or high-resolution peripheral quantitative computed tomography (HR-pQCT).**

| Study | Country | Study design (sample size) | Sample size by group | Obesity criterion | Age (mean ± SD) | Sex (% female) | Assessment tool | Bone site | Bone quality and strength parameters assessed | Quality score[a] |
|---|---|---|---|---|---|---|---|---|---|---|
| **Premenopausal women** | | | | | | | | | | |
| Pollock 2007 | USA | CS (115) | OB: 22 | OB: Body fat≥32% | OB: 18.4 ± 0.5 | 100 | pQCT | Radius, Tibia | Cortical thickness | 8 |
| | | | NO: 93 | NO: 18.2 ± 0.4 | NO: Body fat<32% | | | | | |
| Pollock 2011 | USA | CS (48) | OB: 15 | OB: Body fat≥32% | OB: 19.0 ± 1.1 | 100 | pQCT | Radius, Tibia | Cortical thickness | 6 |
| | | | NO: 33 | NO: 19.3 ± 1.3 | NO: Body fat<32% | | | | | |
| Kassanos 2010 | Greece | Case-control (45) | OB: 15 | OB: BMI≥28 | OB: 28.5 ± 4.1 | 100 | pQCT | Tibia | Cortical thickness | 6 |
| | | | NO: 30 | NO: BMI≤27 | NO: 26.6 ± 5.7 | | | | | |
| **Studies not included in the meta-analysis** | | | | | | | | | | |
| Andersen 2014 | Denmark | CS (72) | OB: 36 | OB: BMI>30 | OB: 41± 8 | 66.7 | HR-pQCT | Radius, Tibia | Cortical thickness, Cortical porosity, Trabecular number, Trabecular separation, Estimated stiffness, Estimated failure load | 7 |
| | | | NO: 36 | NO: BMI 19.5–24.8 | NO: 40.1 ± 7.8 | | | | | |
| Evans 2015 | UK | CS (223) | OB: 146 | OB: BMI≥30 | OB: 49.8 ± 9.9 | 50.7 | HR-pQCT | Radius, Tibia | Cortical thickness, Cortical porosity, Trabecular number, Trabecular separation, Estimated stiffness, Estimated failure load | 8 |
| | | | NO: 77 | NO: BMI 18.5–24.9 | NO: 49.8 ± 9.8 | | | | | |
| Scott 2018 | Australia | CS (168) | OB: 79 | OB: BMI≥30 | 67.7 ± 8.4 | 55.4 | pQCT | Tibia | Cortical thickness | 6 |
| | | | NO: 89 | NO: BMI<30 | | | | | | |
| Sornay-Rendu 2013 | France | Case-control (189) | OB: 63 | OB: BMI≥30 | OB: 68.6 ± 7 | 100 | HR-pQCT | Radius, Tibia | Cortical thickness, Cortical porosity, Trabecular number, Trabecular separation, Estimated stiffness, Estimated failure load | 8 |
| | | | NO: 126 | NO: BMI 18.5–24.9 | NO: 68.2 ± 7.4 | | | | | |

CS: cross-sectional; OB: obese; NO: non-obese; BMI: Body Mass Index; WC: Waist circumference.

BMI is expressed in kg/m$^2$.

WC is expressed in cm.

[a]Quality score was obtained from the Joanna Briggs Institute tool (JBI): <4: high risk of bias; 4–6 moderate risk of bias; ≥7 low risk of bias.

## Results of individual studies

Summary data of individual outcomes for each study are presented using forest plots (**Figs 2–4 and S1–S8**). Results from subgroup analyses for BMD and circulating bone turnover markers outcomes are presented in **S4 Table**.

## Syntheses of results

**Association between obesity and risk of fractures.** *Any fracture.* Fracture data was available in 20 studies [25, 55, 66, 82, 83, 90, 98, 99, 112, 116, 117, 125, 127, 132, 133, 137, 140, 144, 164, 169], totalizing 3,582,437 participants in whom 60,754 fracture events occurred during a mean follow-up of 6.6 years. In the pooled analysis, obesity was associated with a lower risk of fracture in postmenopausal women (n = 12: RR = 0.86, 95% CI: 0.77, 0.97, P = 0.02, $I^2$ = 97%)

A)

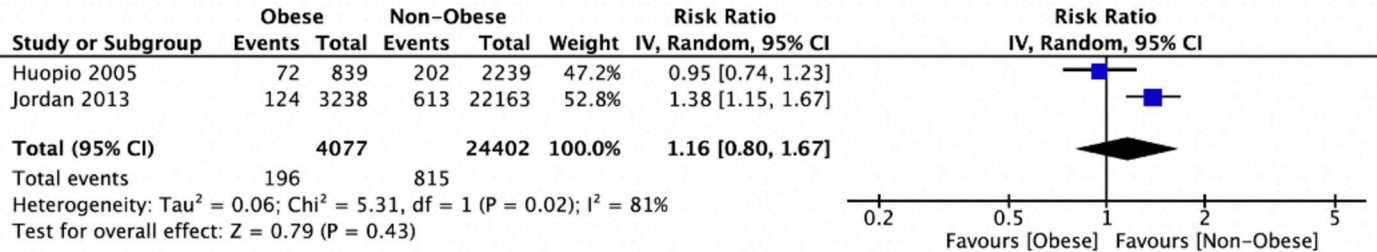

B)

C)

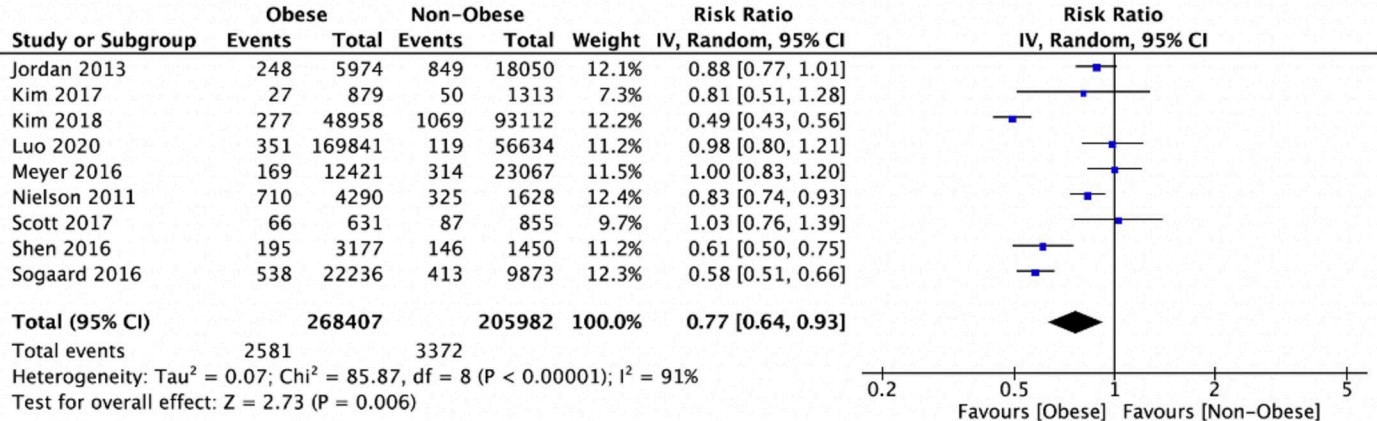

**Fig 2.** Forest plot of pooled effect size for the risk of fracture at any site in A) postmenopausal women, B) premenopausal women, and C) men with vs. without obesity, using a random-effect model.

and men (n = 9: RR = 0.77, 95% CI: 0.64, 0.93, P = 0.006, I² = 91%). No association between obesity and risk of fracture at any site in premenopausal women was found (n = 2: RR = 1.16, 95% CI: 0.80, 1.67, P = 0.43, I² = 81%) (Fig 2). Moreover, there was no association between obesity and risk of fracture in studies combining men and women (n = 4: RR = 0.97, 95% CI: 0.72, 1.31, P = 0.84, I² = 96%). Subgroup analyses did not explain the heterogeneity within groups.

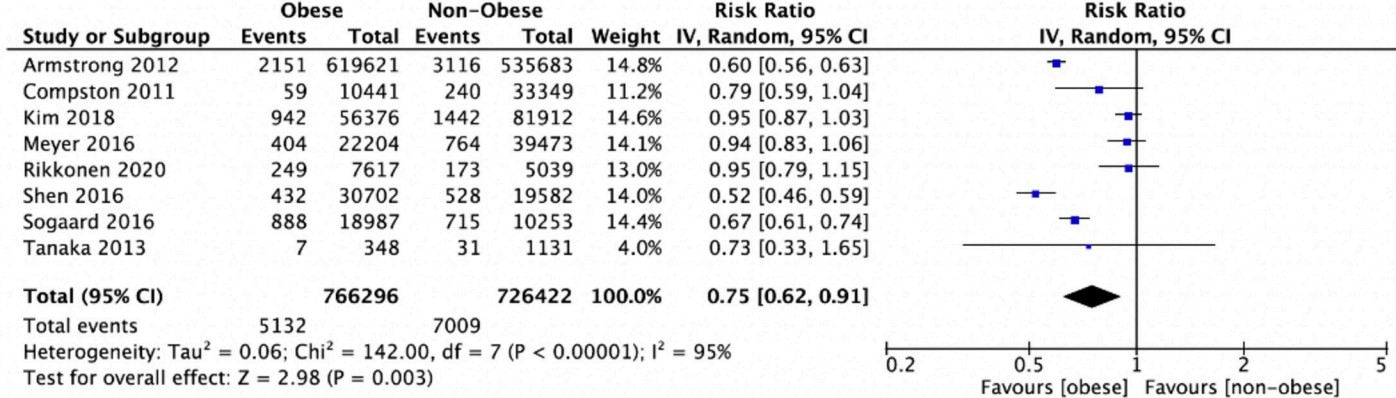

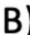

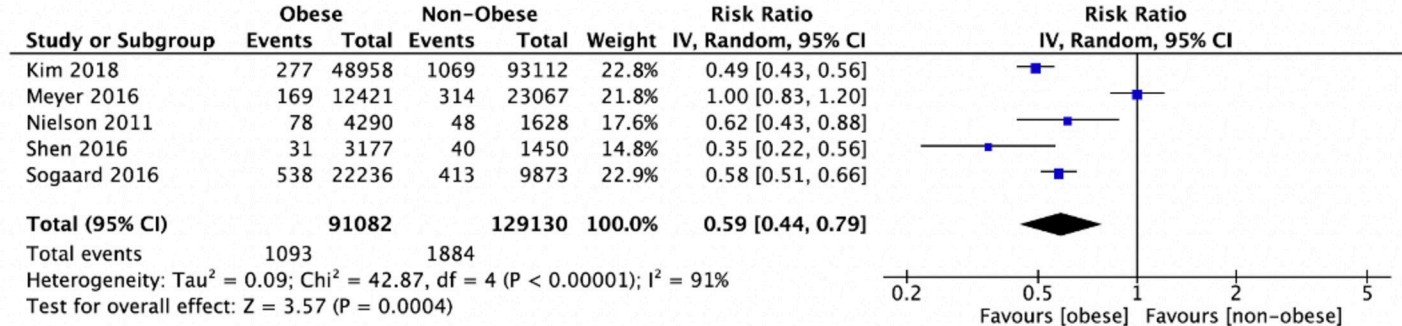

**Fig 3.** Forest plot of pooled effect size for the risk of hip fracture in A) postmenopausal women and B) men with vs. without obesity, using a random-effect model.

*Hip fracture*. Hip fracture data was available in 11 studies [55, 66, 82, 99, 116, 117, 127, 137, 140, 144, 164], including 1,911,715 participants in whom 16,055 fracture events occurred during a mean follow-up length of 7.9 years. Obesity was associated with a lower risk of hip fracture in postmenopausal women (n = 8: RR = 0.75, 95% CI: 0.62, 0.91, P = 0.003, $I^2$ = 95%) and men (n = 5: RR = 0.59, 95% CI: 0.44, 0.79, P = 0.0004, $I^2$ = 91%) (**Fig 3**), but not in studies combining men and women (n = 2: RR = 0.98, 95% CI: 0.55, 1.76, P = 0.96, $I^2$ = 94%). Hip fracture data was not available for studies involving premenopausal women. Subgroup analyses did not explain the heterogeneity within groups.

*Clinical vertebral fracture*. Three studies reported clinical vertebral fractures in postmenopausal women [25, 66, 144], totalizing 315,136 participants in whom 1,694 fracture events occurred during a mean follow-up length of 6.6 years. These studies revealed that obesity was not associated with clinical vertebral fracture risk (**S1 Fig**). Subgroup analyses could not be performed.

*Upper limb fracture*. Two studies reported wrist and forearm fractures [55, 66, 144], including a total of 1,200,573 participants in whom 10,681 fracture events happened during a mean follow-up length of 5.7 years. Studies were conducted in postmenopausal women and showed an association between obesity and a reduced risk of wrist fracture (n = 2: RR = 0.85, 95% CI: 0.81, 0.88, P<0.00001, $I^2$ = 0%) (**S1 Fig**). No difference between groups was observed for forearm fracture (n = 2). Subgroup analyses could not be performed. Meta-analysis could not be performed on humerus fracture since only one included study specifically assessed this site.

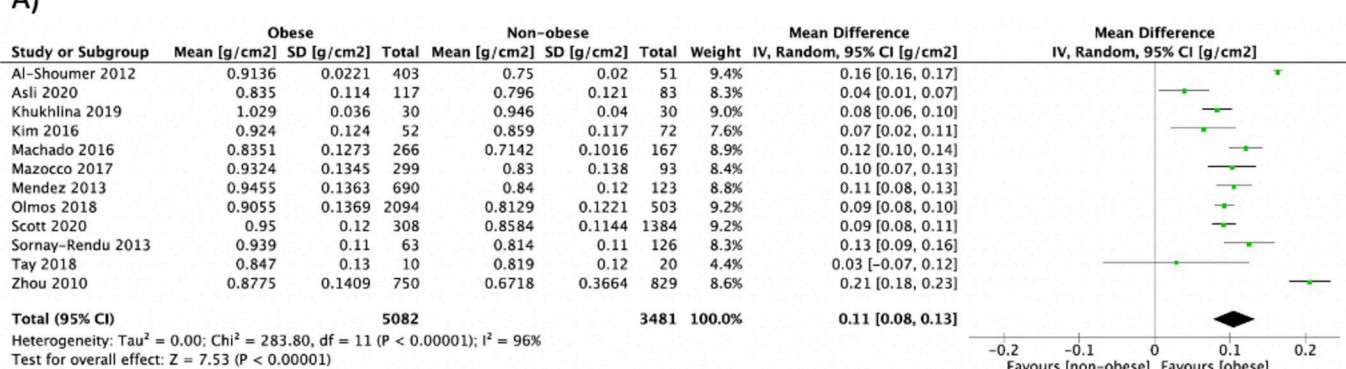

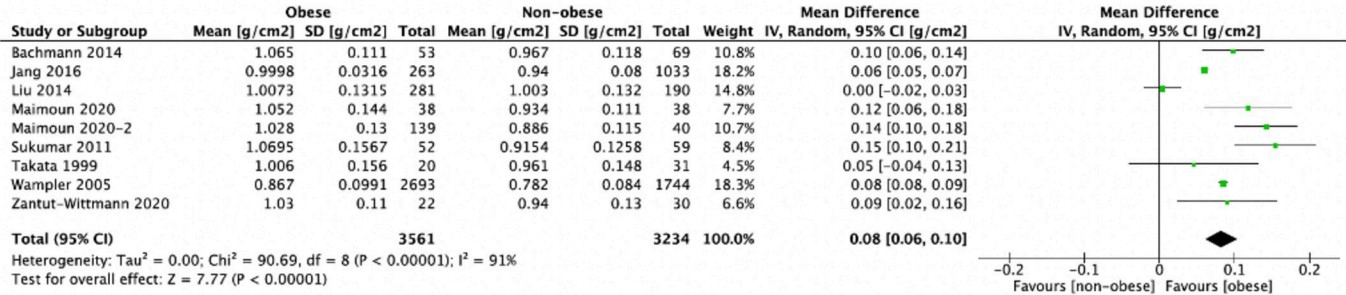

**Fig 4.** Forest plot of pooled effect size for the total hip aBMD by DXA mean difference between A) postmenopausal women, B) premenopausal women and C) men with vs. without obesity, using a random-effect model.

This study reported that high BMI was a risk factor for humerus fracture in postmenopausal women [144].

*Lower limb fracture.* Ankle fracture risk was reported in two studies [55, 66], including 1,198,360 participants in whom 7,221 fracture events arose during a mean follow-up length of 5.4 years. Studies included postmenopausal women and showed that obesity was associated with an increased risk of ankle fracture (RR = 1.60, 95% CI: 1.52, 1.68, P<0.00001, $I^2$ = 0%) (**S1 Fig**). Subgroup analysis could not be performed. Moreover, meta-analysis could not be performed on either tibia/fibula or femur (non-hip) fracture. Yet, one study reported an increased risk of upper leg fracture in postmenopausal women with obesity [17].

**Association between obesity and BMD.** *Total hip aBMD by DXA.* Total hip aBMD by DXA was reported in 33 studies [21, 52, 54, 56, 58, 68, 70, 71, 87, 89, 93, 96, 97, 107, 110–114, 117, 118, 130, 132, 133, 143, 148, 151–153, 163, 173, 178, 179], including 29,279 participants.

Obesity was associated with a higher total hip aBMD in postmenopausal women (n = 12: MD = 0.11 g/cm$^2$, 95% CI: 0.08, 0.13, P<0.00001, I$^2$ = 96%), premenopausal women (n = 9: MD = 0.08 g/cm$^2$, 95% CI: 0.06, 0.10, P<0.00001, I$^2$ = 91%), men (n = 9: MD = 0.07 g/cm$^2$, 95% CI: 0.05, 0.09, P<0.00001, I$^2$ = 82%), and in studies combining men and women (n = 7: MD = 0.09 g/cm$^2$, 95% CI: 0.07, 0.11, P<0.00001, I$^2$ = 77%) (**Fig 4**). Subgroup analyses did not explain the heterogeneity within groups.

*Femoral neck aBMD by DXA*. Femoral neck aBMD by DXA was reported in 48 studies [52, 54, 56, 58–60, 67–72, 84, 85, 88, 89, 92, 93, 96, 97, 101, 103, 106–108, 110–114, 118, 120, 121, 124, 128, 130, 142, 144, 148–153, 167, 173, 177, 180, 181], including 30,577 participants. Obesity was associated with increased femoral neck aBMD in postmenopausal women (n = 21: MD = 0.06 g/cm$^2$, 95% CI: 0.05, 0.08, P<0.00001, I$^2$ = 90%), premenopausal women (n = 13: MD = 0.05 g/cm$^2$, 95% CI: 0.03, 0.07, P<0.00001, I$^2$ = 92%), men (n = 8: MD = 0.05 g/cm$^2$, 95% CI: 0.03, 0.07, P<0.00001, I$^2$ = 79%), and in studies combining men and women (n = 7: MD = 0.07 g/cm$^2$, 95% CI: 0.04, 0.10, P<0.00001, I$^2$ = 77%) (**S2 Fig**). Subgroup analyses did not explain the heterogeneity within groups.

*Lumbar spine aBMD by DXA*. Lumbar spine aBMD measured by DXA was reported in 56 studies [20–22, 52, 54, 56, 59–61, 67–72, 78, 81, 84, 85, 87–89, 92–94, 97, 100, 103, 106–108, 110, 112–115, 118, 120, 121, 124, 126, 128, 130–132, 134, 135, 142–144, 148, 150–153, 163, 167, 173, 177–181], including 29,420 participants. Obesity was associated with increased lumbar spine aBMD in postmenopausal women (n = 27: MD = 0.07 g/cm$^2$, 95% CI: 0.05, 0.09, P<0.00001, I$^2$ = 92%), premenopausal women (n = 17: MD = 0.07 g/cm$^2$, 95% CI: 0.04, 0.09, P<0.0001, I$^2$ = 90%), men (n = 8: MD = 0.06 g/cm$^2$, 95% CI: 0.04, 0.08, P<0.00001, I$^2$ = 48%), and in studies combining men and women (n = 12: MD = 0.06 g/cm$^2$, 95% CI: 0.03, 0.08, P<0.00001, I$^2$ = 93%) (**S3 Fig**). Subgroup analyses did not explain the heterogeneity within groups.

*Radius aBMD by DXA*. Radius aBMD measured by DXA was available in 16 studies [21, 58, 69, 70, 73, 75, 102, 106, 110, 121, 124, 138, 145, 165, 178, 179], including 10,008 participants. Obesity was associated with higher aBMD at the radius in postmenopausal women (n = 6: MD = 0.07 g/cm$^2$, 95% CI: 0.05, 0.08, P<0.00001, I$^2$ = 65%), premenopausal women (n = 10: MD = 0.03 g/cm$^2$, 95% CI: 0.02, 0.04, P<0.00001, I$^2$ = 84%) and men (n = 2: MD = 0.02 g/cm$^2$, 95% CI: 0.01, 0.03, P<0.00001, I$^2$ = 0%) (**S4 Fig**). Subgroup analyses did not explain the heterogeneity within groups.

*Radius volumetric BMD (vBMD) by pQCT and HR-pQCT*. The two studies that reported radius vBMD by pQCT in premenopausal women revealed no difference between those with or without obesity (**S5 Fig**) [122, 123].

*Tibia vBMD by pQCT and HR-pQCT*. Two studies reported tibia vBMD measured by pQCT, which included 331 premenopausal women [122, 123]. Similar to the radius vBMD findings by pQCT, obesity was not associated with any difference in tibia vBMD (**S5 Fig**) [122, 123].

**Associations between obesity, bone microarchitecture and strength.** *Radius cortical thickness by pQCT and HR-pQCT*. Radius cortical thickness by pQCT was reported in two studies [122, 123], which included 163 premenopausal women. Those studies did not reveal any association between radius cortical thickness and obesity (**S6 Fig**).

*Tibia cortical thickness by pQCT and HR-pQCT*. Three studies reported tibia cortical thickness by pQCT [95, 122, 123] in premenopausal women and found no difference between premenopausal women with and without obesity (**S6 Fig**).

*Radius and tibia cortical porosity by HR-pQCT*. Three studies excluded from the meta-analysis reported radius and tibia cortical porosity by HR-pQCT [20–22]. At both sites, cortical porosity was lower in postmenopausal women with obesity compared to women without

obesity [21]. Another study revealed significantly lower cortical porosity at the tibia in men aged 55–75 years and postmenopausal women with obesity, whereas no significant difference was observed at the radius [22]. In the third study, cortical porosity at the radius and tibia was not different between individuals with or without obesity in a mixed population of men and women (mean age 41 years, 66.7% women) [20].

*Radius and tibia trabecular number and trabecular separation by HR-pQCT.* The same studies reported radius and tibia trabecular number and trabecular separation by HR-pQCT [20–22]. Radius trabecular number was significantly greater in individuals with obesity in all studies, whereas radius trabecular separation was significantly lower in postmenopausal women [21, 22], men and premenopausal women with obesity [22], compared controls without obesity. Moreover, tibia trabecular number was significantly greater, and trabecular separation was significantly lower in men [22], pre- and postmenopausal women [21, 22], and in a mixed population of men and women with obesity (mean age 41 years, 66.7% women) [20].

*Radius and tibia estimated stiffness and failure load by HR-pQCT.* The same studies also reported radius and tibia estimated stiffness and failure load by HR-pQCT [20–22]. At the radius, the estimated stiffness was higher in postmenopausal women [21, 22] and men aged 55–75 years with obesity [22], whereas no difference was observed in premenopausal women and in younger men aged 25–40 years [22]. Nevertheless, the estimated failure load at the radius was greater for men [22], pre- and postmenopausal women with obesity [21, 22]. At the tibia, both the estimated stiffness and failure load were higher in postmenopausal women [21, 22], premenopausal women and men with obesity [22]. However, the study conducted in a mixed population of men and women found no difference between individuals with and without obesity for both the radius and tibia estimated stiffness and failure load (mean age 41 years, 66.7% women) [20].

**Association between obesity and circulating bone turnover markers.** *P1NP levels.* P1NP levels were reported in 13 studies [21, 22, 64, 70, 88, 104, 112, 118, 129, 139, 146, 147, 153], including 5,808 participants. Obesity was associated with lower P1NP levels in studies combining men and women (n = 5: MD = -7.66 ng/ml, 95% CI: -13.36, -1.96, P = 0.008, $I^2$ = 68%), but not in postmenopausal women (n = 8) (**S7 Fig**). Subgroup analyses did not explain the heterogeneity within groups.

*Total osteocalcin levels.* Total osteocalcin levels were reported in 29 studies [21, 53, 60, 62–65, 68, 74, 76, 77, 80, 81, 84, 92, 97, 105, 107, 119, 126, 129, 135, 144, 146, 147, 151, 153, 166, 175], including 6,332 participants. Obesity was not associated with any difference in osteocalcin levels between individuals with and without obesity (**S7 Fig**), except in studies combining men and women (n = 9: MD = -3.86 ng/ml, 95% CI: -6.78, -0.95, P = 0.009, $I^2$ = 97%). Subgroup analyses did not explain the heterogeneity within groups.

*CTX levels.* CTX levels were reported in 21 studies [21, 22, 60, 63, 64, 68, 70, 81, 86, 88, 91, 97, 104, 107, 112, 118, 129, 139, 146, 147, 171], including 10,375 participants. Obesity was associated with reduced CTX levels in postmenopausal women (n = 12: MD = -0.08 ng/ml, 95% CI: -0.12, -0.04, P<0.0001, $I^2$ = 75%) (**S8 Fig**) and in studies combining men and women (n = 9: MD = -0.08 ng/ml, 95% CI: -0.12, -0.04, P<0.0001, $I^2$ = 74%). Subgroup analyses did not explain the heterogeneity within groups.

*Urinary NTX levels.* Urinary NTX levels were reported in 5 studies [79, 135, 144, 153, 182], including 3,329 participants. No difference between individuals with and without obesity was observed in postmenopausal women (n = 3) (**S8 Fig**) and in studies with a mixed population (n = 2). No subgroup analyses were performed.

*Sclerostin levels.* Sclerostin levels were reported in 3 studies [53, 57, 79], including 380 participants. In those studies, no difference between individuals with and without obesity was observed. No subgroup analyses were performed.

### Risk of bias across studies and quality of evidence

Strong evidence of heterogeneity was observed between studies for the majority of the outcomes. Publication bias for all outcomes were assessed using funnel plots (**S9–S19 Figs**). We saw no evidence of asymmetry; therefore, no publication bias was detected. Publication bias could only be assessed for the outcomes that had a sufficient sample size [49]: fracture at any site in postmenopausal women and men, hip fracture in postmenopausal women, total hip aBMD in postmenopausal women, femoral neck aBMD in postmenopausal and premenopausal women, lumbar spine aBMD in postmenopausal women, premenopausal women and in studies with a mixed population of men and women, osteocalcin levels and CTX levels in postmenopausal women. The quality of evidence assessed following the GRADE approach was considered very low for all fracture outcomes except for wrist fracture in postmenopausal women, where the quality of evidence was considered low. The quality of evidence was also considered low for lumbar spine aBMD in men, radius and tibia vBMD by pQCT, radius and tibia cortical thickness by pQCT, and P1NP levels in premenopausal women. The quality of evidence was considered very low for all other outcomes. Of note, the quality of evidence was downgraded mainly because of the study design of included studies (which were not randomized controlled trials) and the inconsistency in results.

### Heterogeneity exploration

When studies were removed from the analysis one at a time, we found one study [111] that had a strong effect on the heterogeneity for total hip aBMD in a mixed population of men and women. Indeed, we found that the study by Lloyd *et al.* [111] was responsible for the majority of the heterogeneity. When this study was removed from the pooled estimate, the Higgin's $I^2$ decreased from 80% to 1% and the pooled mean difference decreased from 0.09 to 0.08 g/cm$^2$ (95% CI: 0.07, 0.09, P<0.00001). Even if the study by Lloyd *et al.* [111] was the main source of heterogeneity for this outcome, we decided to maintain this study in the analyses since it was not significantly affecting the pooled estimate, had a group with and without obesity with a similar proportion of men and women with comparable age, and a low risk of bias. However, potential explanation for the observed heterogeneity may be the higher prevalence of diabetes and proportion of black individuals in the group with obesity compared with the group without obesity, which are both known to be associated with higher BMD [183, 184]. Heterogeneity exploration was performed for all outcomes. However, no other study was found to have a strong effect on heterogeneity.

## Discussion

### Summary of evidence

One hundred and thirty-four studies totalizing more than 5 million individuals were included in this systematic review, of which 121 studies were incorporated in the meta-analysis. Our results showed a significantly reduced risk of fracture in postmenopausal women and men with obesity compared with individuals without obesity. Assessment of fracture risk by anatomical site revealed that postmenopausal women with obesity had a lower risk of hip and wrist fracture by 25% and 15%, respectively, whilst ankle fracture risk was increased by 1.6-fold compared with postmenopausal women without obesity. Hip fracture risk was reduced by 41% in men with vs. without obesity. Finally, obesity was not associated with clinical vertebral fracture risk, but only a handful of studies assessed this outcome specifically, and it is not clear if ascertainment was complete in these studies. These results confirm that fracture risk varies by skeletal site in individuals with obesity, and also suggests that the impact of

obesity on fracture differs in men and postmenopausal women. No conclusion could be drawn regarding the association between obesity and fracture incidence in premenopausal women given the small number of studies. Moreover, the impact of combined obesity and type 2 diabetes on fracture risk could not be assessed, as no study specifically addressed this question. High heterogeneity was observed between studies for most outcomes, which was not fully explained in subgroup or sensitivity analyses. Lastly, the overall quality of evidence based on the GRADE approach was very low to low for all outcomes, due to the study designs and risk of bias of the included studies, and the high heterogeneity between studies.

Regarding BMD and bone microarchitecture, the available evidence suggests favorable findings in people with obesity vs. controls without obesity. Indeed, aBMD by DXA was higher at the total hip, femoral neck, lumbar spine and radius in men, premenopausal women and postmenopausal women with obesity compared with their counterpart without obesity. Only two studies conducted in postmenopausal women as well as in premenopausal women and men found superior HR-pQCT-derived bone microarchitecture and strength in individuals with obesity compared with controls without obesity: tibia vBMD was greater, radius cortical thickness was higher, radius and tibia trabecular number were increased, trabecular separation was reduced, and estimated stiffness and failure load were increased. Finally, the bone resorption marker CTX was generally lower in people with obesity. However, conflicting results were reported for the bone formation markers P1NP and osteocalcin, with either no difference or lower levels in those with vs without obesity. In a limited number of studies, no difference between groups was observed in the osteocyte marker sclerostin. To the best of our knowledge, our meta-analysis is the first to evaluate, altogether, the relationship between obesity, fracture risk, BMD and bone quality parameters by sex and menopausal status.

Our finding of a decreased risk of hip fracture in men and postmenopausal women with obesity is consistent with a previous meta-analysis, which reported that high BMI is a protective factor for hip fracture in postmenopausal women [18], as well as in men and women of all age [29]. This fracture risk reduction is clinically significant since hip fractures are associated with the highest morbidity and mortality rates [1, 185], and impose a financial burden on society [7]. However, opposite to our results, another meta-analysis found that abdominal obesity is associated with a higher risk of hip fracture in men and women aged 40 years and older [30]. These conflicting results may be explained by the fact that the majority of the studies included in our meta-analysis and previous meta-analyses focused on general obesity, mostly defined by BMI, rather than abdominal obesity. While abdominal obesity has been recognized as a stronger risk factor of metabolic disorders than BMI, this may also be the case for bone fragility [186, 187]. Abdominal obesity is associated with greater insulin resistance as well as systemic inflammation and oxidative stress [188, 189], increased circulating inflammatory cytokines, and altered levels of bone-regulating hormones [190], which are all known to adversely affect bone metabolism. Moreover, using BMI as a measure of adiposity has been shown to be less accurate in older adults due to change in body composition associated with aging [191]. Altogether, those with abdominal obesity may have a distinct fracture risk pattern, highlighting the necessity to consider abdominal obesity when assessing fracture risk in adults [25].

In addition, type 2 diabetes, which frequently coexists with obesity, may further impact fracture risk. Indeed, many studies reported increased risk of hip and non-vertebral fracture in individuals with type 2 diabetes [19, 192]. However, studies considering presence of type 2 diabetes in the association between obesity and fracture risk are limited: most studies used type 2 diabetes as an adjustment factor and did not assess whether the presence of type 2 diabetes modifies the association between obesity and fracture incidence.

Our meta-analysis supports that the association between obesity and risk of fracture is skeletal site-specific. This is also supported by another meta-analysis which found that obesity was

a risk factor of lower limb fracture and upper arm fracture (humerus and elbow) in women of all age [18]. Reasons for this site-specific association are still not completely understood, but it appears that specific bone sites may require enhancement of different material properties to resist fracture depending on the predominant failure mechanism at that site [193]. Thus, the hip and wrist in individuals with obesity may be more protected from fracture due to the increased BMD which improves bone strength, while sites such as the vertebrae or lower limbs fracture via other failure mechanisms, which require enhancement in other material properties (i.e. fatigue strength and fracture toughness). Although individuals with obesity are more likely to fall due to reduced mobility, postural control and protective responses [194, 195], and even weakened psychomotor abilities [196], soft tissue padding around the hip area may allow energy dissipation after trauma or a fall, subsequently contributing to the protective effect of obesity against hip fracture [197]. Moreover, a different falls pattern may exist between individuals with and without obesity, as individuals with obesity are more likely to fall backward or sideways, rather than forward [196]. Therefore, wrists are less exposed to trauma, which may explain the reduced risk of fracture at this site. Another possible explanation is that ankles are not protected by adipose tissue padding, and have to support greater body weight when falling, perhaps explaining the increased risk of fracture at these sites. Besides, higher body weight increases the impact forces during the fall.

Another goal of this meta-analysis was to evaluate differences in BMD, bone microarchitecture and bone remodeling markers between adults with and without obesity to help understand the bone parameters involved in the obesity-associated bone fragility. To our knowledge, this is the first meta-analysis to address and quantify the differences in BMD and bone quality parameters in this population. Our results showed that overall, individuals with obesity have higher aBMD, vBMD (when assessed by HR-pQCT) and better bone microarchitecture and strength at all sites. However, conflicting results remain for cortical porosity, since either lower or similar cortical porosity was observed between individuals with and without obesity at both sites. In all studies, cortical porosity was measured using a first-generation HR-pQCT scanner, which limited the measurement to peripheral rather than diaphyseal sites of the radius and tibia, and it is known that cortical porosity has poor precision [198]. Moreover, one of the studies excluded participants with type 2 diabetes [22] whereas the other two studies did not mention the diabetes status of the population [20, 21]. Since cortical porosity has been shown to be increased in individuals with type 2 diabetes but decreased in obesity, it may explain, at least partly, the conflicting results for this outcome. Nevertheless, it is unknown whether the increase in BMD and bone strength as well as favorable bone microarchitecture seen in individuals with obesity is sufficient to resist the larger strains applied on bones during trauma or fall in the context of excess weight. Besides, it is difficult to make any definitive conclusions since only a limited number of studies compared these bone quality parameters using advanced imaging techniques in individuals with or without obesity. Finally, with regards to circulating bone turnover markers, our meta-analysis revealed significantly lower levels of the bone resorption marker CTX in individuals with obesity but results on the bone formation markers P1NP and osteocalcin were mixed. Potential causes for these inconsistent results are the heterogeneity of the populations included (i.e. diabetes status) and the preanalytical and analytical variability of the bone turnover markers measurements (i.e. fasting status and time of day of the measurement, measurement in serum or plasma, analysis in a single batch or not, type of assay).

## Limitations and strengths

Our systematic review and meta-analysis has strengths, such as the exhaustive search strategy and number of outcomes investigated. Indeed, it included 134 studies, which allowed us to

highlight the magnitude of the association between obesity and risk of any and site-specific fracture, and the difference in BMD, bone microarchitecture parameters and circulating bone remodeling markers between individuals with and without obesity, stratified by sex and menopausal status. The quality of all included studies was also assessed with validated quality assessment tools for cross-sectional, cohort and case-control studies. We carried out an extensive quality assessment for individual studies and for each outcome using the GRADE approach. We also investigated heterogeneity with subgroup analyses and performed sensitivity analyses.

Our meta-analysis has also limitations. First, conclusions could not be drawn with regards to fracture incidence in premenopausal women, in men (except for hip fracture), and for humerus, tibia/fibula and femur (non-hip) fracture incidence in postmenopausal women. Second, high heterogeneity was observed between the included studies, which was not totally explained in subgroup analyses. The inclusion of studies using a cut-off of 70% of men and pre- and postmenopausal women to categorize groups by sex and menopausal status may have increased heterogeneity within groups. Heterogeneity may also be the result of the combination of obese with overweight individuals in some studies as well as of a wide range of BMI across studies. Unfortunately, we could not perform subgroup analyses based on BMI categories, as very few studies classified the obese group based on BMI obesity categories. Moreover, very few studies considered a different criterion for obesity than BMI, which does not necessarily follow the dose-response relationship between obesity and fracture risk. Therefore, using BMI as a criterion does not discriminate individuals who are at higher risk vs lower risk of fracture. Remaining heterogeneity may be related, at least partly, to the demographic diversity of the populations across studies (i.e., ethnicity, age and socioeconomic level), the presence of conditions or use of certain medications that may affect bone outcomes for some individuals (e.g. diabetes status), and the method used to report fractures (adjudicated or self-reported). Also, for fracture outcomes, adjustment for covariates and lengths of follow-up were not consistent across studies, and mechanism of fracture was not always reported (fragility vs. non-fragility fracture). Third, risk of vertebral fractures may have been underestimated since only clinical vertebral fractures were reported. Fourth, while type 2 diabetes often coexists with obesity and may further impair bone quality and reduce bone strength in this population, we have not been able to examine the association between obesity, with and without type 2 diabetes, on bone outcomes. Indeed, most studies only reported prevalence of participants with type 2 diabetes and used it as an adjustment factor in the statistical analyses. Fifth, only a few studies compared bone microarchitecture parameters in people with or without obesity. Finally, the inclusiveness of our analysis may be limited by the fact that studies reporting correlation analyses or relative or absolute measures of effect without the number of fracture events were not included.

## Conclusions

In conclusion, we found that obesity is associated with higher bone mass and favorable bone microarchitecture while bone turnover, as assessed by circulating bone turnover markers, was either lower or similar to controls without obesity. Obesity was associated with a lower risk of fracture at the hip (in men and postmenopausal women) and at the wrist (in postmenopausal women) but with a higher risk of ankle fracture (in postmenopausal women). Results should however be interpreted with caution given the high heterogeneity among studies for most outcomes, and the low quality of evidence for all outcomes. Moreover, no conclusion could be drawn for premenopausal women and for certain fracture sites in all groups given the paucity of data. This meta-analysis highlights areas for future research including the need for site-specific fracture studies in premenopausal women with obesity, studies evaluating fracture sites

other than the hip in men with obesity or comparing bone microarchitecture between pre- and postmenopausal women as well as men with and without obesity. It also emphasizes the need to standardize the assessment of bone turnover markers in research. Moreover, studies looking at the impact of fat distribution on bone outcomes may find obesity patterns that may be more susceptible to bone fragility, as defining obesity with BMI may not be specific enough to portray bone metabolism impairment in individuals with obesity. Finally, as type 2 diabetes often coexists with obesity and is a well-known risk factor for fracture, studies addressing specifically the impact of type 2 diabetes in this population are necessary.

## Supporting information

**S1 Checklist. PRISMA 2009 checklist.**
(DOC)

**S1 Table. Search strategy.**
(DOCX)

**S2 Table. Study characteristics of included studies for bone turnover markers outcome.**
(DOCX)

**S3 Table. Assessment methods used for bone turnover markers.**
(DOCX)

**S4 Table. Results of subgroup analysis by obesity and risk of bias criterion for bone mineral density and bone turnover markers outcomes in postmenopausal women, premenopausal women and men.**
(DOCX)

**S1 Fig.** Forest plot of pooled effect size for the risk of A) clinical vertebral fracture, B) wrist fracture, C) forearm fracture and D) ankle fracture in postmenopausal women with vs. without obesity, using a random-effect model.
(DOCX)

**S2 Fig.** Forest plot of pooled effect size for the femoral neck aBMD by DXA mean difference between A) postmenopausal women, B) premenopausal women and C) men with vs. without obesity, using a random-effect model.
(DOCX)

**S3 Fig.** Forest plot of pooled effect size for the lumbar spine aBMD by DXA mean difference between A) postmenopausal women, B) premenopausal women and C) men with vs. without obesity, using a random-effect model.
(DOCX)

**S4 Fig.** Forest plot of pooled effect size for the radius aBMD by DXA mean difference between A) postmenopausal women, B) premenopausal women and C) men with vs. without obesity, using a random-effect model.
(DOCX)

**S5 Fig.** Forest plot of pooled effect size for the A) radius vBMD and B) tibia vBMD by pQCT mean difference between premenopausal women with vs. without obesity, using a random-effect model.
(DOCX)

**S6 Fig.** Forest plot of pooled effect size for the A) radius cortical thickness and B) tibia cortical thickness by pQCT mean difference between premenopausal women with vs. without obesity,

using a random-effect model.
(DOCX)

**S7 Fig.** Forest plot of pooled effect size for A) P1NP levels mean difference between postmenopausal women with vs. without obesity, and total osteocalcin levels mean difference between B) postmenopausal women, C) premenopausal women and D) men with vs. without obesity, using a random-effect model.
(DOCX)

**S8 Fig.** Forest plot of pooled effect size for A) CTX levels and B) NTX levels mean difference between postmenopausal women with vs. without obesity, using a random-effect model.
(DOCX)

**S9 Fig. Funnel plot for fracture at any site in postmenopausal women.**
(DOCX)

**S10 Fig. Funnel plot for fracture at any site in men.**
(DOCX)

**S11 Fig. Funnel plot for hip fracture in postmenopausal women.**
(DOCX)

**S12 Fig. Funnel plot for total hip aBMD in postmenopausal women.**
(DOCX)

**S13 Fig. Funnel plot for femoral neck aBMD in postmenopausal women.**
(DOCX)

**S14 Fig. Funnel plot for femoral neck aBMD in premenopausal women.**
(DOCX)

**S15 Fig. Funnel plot for lumbar spine aBMD in postmenopausal women.**
(DOCX)

**S16 Fig. Funnel plot for lumbar spine aBMD in premenopausal women.**
(DOCX)

**S17 Fig. Funnel plot for lumbar spine aBMD in studies combining men and women.**
(DOCX)

**S18 Fig. Funnel plot for osteocalcin levels in postmenopausal women.**
(DOCX)

**S19 Fig. Funnel plot for CTX levels in postmenopausal women.**
(DOCX)

## Author Contributions

**Conceptualization:** Anne-Frédérique Turcotte, Sarah O'Connor, Suzanne N. Morin, Jenna C. Gibbs, Bettina M. Willie, Sonia Jean, Claudia Gagnon.

**Data curation:** Anne-Frédérique Turcotte, Sarah O'Connor.

**Formal analysis:** Anne-Frédérique Turcotte.

**Methodology:** Anne-Frédérique Turcotte, Sarah O'Connor, Suzanne N. Morin, Jenna C. Gibbs, Bettina M. Willie, Sonia Jean, Claudia Gagnon.

**Supervision:** Sonia Jean, Claudia Gagnon.

**Validation:** Anne-Frédérique Turcotte, Sarah O'Connor.

**Writing – original draft:** Anne-Frédérique Turcotte.

**Writing – review & editing:** Anne-Frédérique Turcotte, Sarah O'Connor, Suzanne N. Morin, Jenna C. Gibbs, Bettina M. Willie, Sonia Jean, Claudia Gagnon.

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
