## [Decision Letter · Decision Letter 0]

1 Mar 2021

PONE-D-21-02000

Effects of Obesity on Risk of Fracture, Bone Mineral Density and Bone Quality in Adults: A systematic Review and Meta-analysis.

PLOS ONE

Dear Dr. Gagnon,

Thank you for submitting your manuscript to PLOS ONE. After careful consideration, we feel that it has merit but does not fully meet PLOS ONE’s publication criteria as it currently stands. Therefore, we invite you to submit a revised version of the manuscript that addresses the points raised during the review process. 

We look forward to receiving your revised manuscript.

Kind regards,

Tuan V. Nguyen

Academic Editor

PLOS ONE

Journal Requirements:

Additional Editor Comments:

Thank you for submitting the manuscript 'Effects of obesity on risk of fracture ...' for consideration for publication in PLoS ONE. Your manuscript has now been reviewed by 2 experts, and their comments are attached for your perusal. As you will see, both reviewers recognize the importance of your work, but they also raise a number of issues concerning methodology and interpretation. I invite you to comment on their concerns.

As an Academic Editor, I have read your manuscript with interest. I also think that your manuscript has merit, but I would like to take care of the following points:

1. I am concerned about inclusiveness of your analysis. I understand that the relationship between BMI and fracture or BMD has been examined by many studies around the world, and some of the studies that I am familiar with were not included in your analysis. For instance, the study by Chan et al (https://pubmed.ncbi.nlm.nih.gov/24862213/) was not included in your analysis.

2. The BMI threshold for defining 'obesity' is different across countries/populations. How did you account for this differences in your analysis?

3. Your conclusion is not clear at all. Readers (and I) want to now what is the substantive message you want to convey. It appears to me that your data show that obesity was associated with higher bone mass, bone quality, and lower risk of fracture. Please consider rewording your conclusion to be consistent with the data.

4. The title: I consider that the word 'effect' is not quite appropriate for this manuscript, because all studies were either cross-sectional or cohort investigations that can only delineate an association, not effect. Please consider another title.

Reviewers' comments:

Reviewer's Responses to Questions

**Comments to the Author**

1. Is the manuscript technically sound, and do the data support the conclusions?

Reviewer #1: Yes

Reviewer #2: Yes

2. Has the statistical analysis been performed appropriately and rigorously? 

Reviewer #1: Yes

Reviewer #2: Yes

3. Have the authors made all data underlying the findings in their manuscript fully available?

Reviewer #1: Yes

Reviewer #2: Yes

4. Is the manuscript presented in an intelligible fashion and written in standard English?

Reviewer #1: Yes

Reviewer #2: Yes

5. Review Comments to the Author

Reviewer #1: I congratulate the authors on a well-conducted and timely meta-analysis on the topic of obesity, fractures and bone health. I have few suggestions for improvement but one significant one: there is emerging evidence that obesity, when defined by direct assessments such as body fat measured by DXA, offers little protection for fracture. This suggests that the apparent protective effect of high BMI is explained by higher muscle mass rather than higher fat mass in obese individuals. Is it possible to perform a sensitivity analysis using only studies that defined obesity using measures other than BMI, to explore this potential association?

My remaining comments are generally minor:

1. You used a cut-off of 80% of participants aged 18or older to determine whether studies included adults, and a cut-off of 70% of either sex to determine whether a study included men or women. Can you provide some rationale for these cut-offs?

2. The description of the exposure and comparator groups (pages 6-7) is a little confusing regarding which group included overweight participants (i.e.e BMI range of 25-30). From further reading, it appears overweight participants were allocated to the obese group, but it would be useful to the reader to specify this.

3. "Contrary to what was initially planned, only studies in English or French were considered"... (Page 8). Can you explain what was originally planned and why a change was made?

4. "Finally, 121 studies were included in the meta-analysis(20-22, 25, 51-152, 162-166, 168, 170-172, 174, 176-180): 13(153-161, 167, 169, 173, 175) were excluded because data was missing or could not be transformed"... I assume they were also excluuded because data could not be obtained from authors? If so, this should be specified.

5. You discuss T2DM and how that may have influenced your results in the discussion, but it may be worthwhile specifically mentioning cortical porosity given this appeared to be lower in those with obesity but has previously been shown to be increased in T2DM.

6. Paragraph 2 on page 35 essentially repeats the results and then itself is largely repeated on page 38; this could be removed or merged into the later section on BMD and bone microarchitecture.

Reviewer #2: Turcotte and colleagues conducted a systematic review and meta-analysis on studies collected from PubMed (MEDLINE), EMBASE, Cochrane Library and Web of Science to investigate association between obesity and fracture risk (overall and by site), BMD, and bone quality parameters. Using random-effect model, they found that, compared with ones without obesity, postmenopausal women with obesity had risks of hip and wrist fracture reduced by 25% and 15%, and ankle fracture risk increased by 60%. In men with obesity, hip fracture risk decreased by 41%. Obesity was also associated with increased BMD, better bone microarchitecture and strength, and generally lower or unchanged circulating bone resorption, formation and osteocyte markers. However, the pooled data was based on the original studies' definitions of obesity, of which the cut-offs of definitions varies between studies. The high heterogeneity among studies and overall very low quality of evidence for most outcomes raised the need for further studies in depth.

Although the findings are interesting and meaningful, there are still work to be done:

1. Although documents are provided in somewhere else, it is better to briefly and clearly describe the protocol, especially search strategy, in the text for future readers.

2. It would be better to shorten the methods section, focus on objectives, and mention information in an easy-understanding order. For example, purpose and search strategy first, following by study selection, data extraction and quality assessment, and statistical analyses.

3. In your protocol document registered with PROSPERO, there was no restriction on languages. Why did you exclude studies not in French or English in this study?

4. The key metrics used in data analysis should be defined/explained and with formula if necessary. For instance: kappa static, funnel plot, inverse-variance, and quality of evidence. I2 statistic should be mentioned in bracket after kappa static at the first time for later use. GRADE approach needs to be briefly described, not only cited.

5. Summary measures (Page 12): Choice of meta-analysis method is based on actual value of kappa static. It could be fixed-effects models (homogeneity, kappa ≤50%) or random-effects models (heterogeneity, kappa > 50%).

6. Figure 1: Please include reason for excluding 8,914 records in the screening stage.

7. Page 13: All articles included in or excluded from the research should not be cited in the text. They should be listed in a separate document as a supplementary file.

8. Page 26, 27: Please provide RR, 95% CI, p, and kappa, even there was no association between obesity and risk of fracture in studies combining men and women.

9. Page 27: Several analysis showed low kappa (0%), it would be more appropriate to use fixed-effects model rather than random-effect models.

10. Please provide funnel plot generated in the publication bias assessment to support your points in the result section.

11. Heterogeneity exploration: Please provide results for all outcomes, not only hip BMD.

12. Discussion: Did authors consider the quality of the studies used in pool data? The low evidence of quality in the average results across the studies might be due to their different quality and confounding adjustments.

6. PLOS authors have the option to publish the peer review history of their article (what does this mean?). If published, this will include your full peer review and any attached files.

Reviewer #1: No

Reviewer #2: No

---

## [Author Response · Author response to Decision Letter 0]

9 Apr 2021

Academic Editor

I have read your manuscript with interest. I also think that your manuscript has merit, but I would like you to take care of the following points:

1. I am concerned about inclusiveness of your analysis. I understand that the relationship between BMI and fracture or BMD has been examined by many studies around the world, and some of the studies that I am familiar with were not included in your analysis. For instance, the study by Chan et al (https://pubmed.ncbi.nlm.nih.gov/24862213/) was not included in your analysis.

We thank you for this comment. When we wrote and published the protocol for this systematic review and meta-analysis, we decided to include only the studies that reported fracture incidence (i.e. number of fracture events) per group to allow us to compute relative risks in the meta-analysis. Studies that reported only correlations or relative/absolute measures of effect, such as in the study by Chan et al., were not included since they did not report the number of fracture events (or absolute BMD values) in each group of men and women with normal weight, overweight or obesity. We recognize that this decision limits the inclusiveness of our analysis; we thus added this limitation in the discussion on p.41.

2. The BMI threshold for defining 'obesity' is different across countries/populations. How did you account for these differences in your analysis?

We used the BMI categorization used by the authors to define obesity, which may indeed vary based on ethnicity. Using the author’s definition enabled us to combine studies across countries/populations in the meta-analysis. To help the reader, we reported the obesity criterion used by each author in Tables 1-3 and Supplementary Table 2. While most studies used BMI thresholds to define obesity, a few studies used other measures such as waist circumference or percent body fat. The number of studies using an obesity criterion other than BMI was however too small to perform sensitivity analyses based on the obesity criterion. We added this in the additional analyses section on p.13. 

3. Your conclusion is not clear at all. Readers (and I) want to know what is the substantive message you want to convey. It appears to me that your data show that obesity was associated with higher bone mass, bone quality, and lower risk of fracture. Please consider rewording your conclusion to be consistent with the data.

We have modified the conclusion to ensure the message is clearer. We however did not feel that we could simply state that bone quality was higher and fracture risk was lower. 

Indeed, while bone microarchitecture was favorable in those with obesity, bone turnover, as assessed by circulating bone turnover markers, was either lower (which is not necessarily good) or similar to controls without obesity. Moreover, while hip and wrist fracture risks were lower, ankle fracture risk was higher in postmenopausal women with obesity compared with controls without obesity. Noteworthy, these findings apply only to postmenopausal women and men (for the hip) as there was a lack of fracture data for premenopausal women and men (other than at the hip). Moreover, based on the quality of evidence from GRADE, most of the outcomes had a low or very low quality of evidence mainly due to the study design of included studies and the inconsistency in results (p.33). This is why we wanted to remain careful about our conclusions. We hope that the new formulation conveys a message that is simpler but still reflects adequately our study findings.

4. The title: I consider that the word 'effect' is not quite appropriate for this manuscript, because all studies were either cross-sectional or cohort investigations that can only delineate an association, not effect. Please consider another title.

Thanks for pointing that out. We replaced the word “effect” for “association” in the title.

Reviewer #1

I congratulate the authors on a well-conducted and timely meta-analysis on the topic of obesity, fractures and bone health. 

1. There is emerging evidence that obesity, when defined by direct assessments such as body fat measured by DXA, offers little protection for fracture. This suggests that the apparent protective effect of high BMI is explained by higher muscle mass rather than higher fat mass in obese individuals. Is it possible to perform a sensitivity analysis using only studies that defined obesity using measures other than BMI, to explore this potential association?

We thank you for this comment. In line with your suggestion, we wanted initially to perform a sensitivity analysis based on the obesity criterion used (BMI or on another measure of obesity). However, the number of studies using a measure of obesity other than BMI was too small to allow us to perform this analysis. As you can see in Table 1, only 3 studies out of 20 reporting fracture outcomes used either waist circumference or percent body fat to define obesity, and the same proportions are observed for the other outcomes. We added this in the additional analyses section on p.13. 

My remaining comments are generally minor:

2. You used a cut-off of 80% of participants aged 18 or older to determine whether studies included adults, and a cut-off of 70% of either sex to determine whether a study included men or women. Can you provide some rationale for these cut-offs?

The cut-off of 80% of participants aged 18 or older to determine whether studies included adults is a common arbitrary threshold used in systematic reviews to manage studies with heterogenous study populations i.e., studies including both adults and children. Although this method has its limits, it enables the inclusion of studies with a majority of adults that would have been excluded otherwise. We have added this information in the methods section, on p.6. 

Regarding the grouping of studies based on sex and menopausal status, we used once again an arbitrary threshold of 70% to determine whether a study was included in the men, postmenopausal women or premenopausal women groups. This arbitrary cut-off was chosen to minimise heterogeneity while maximizing statistical power within each group. Due to higher heterogeneity, it is more difficult to draw conclusions for the “mixed population” group as men and women of a wide range of age are mixed. We added the rationale for this cut-off on p.12. We have also added the limitations of this categorization in the “limitations and strengths” paragraph on p.40. 

2. The description of the exposure and comparator groups (pages 6-7) is a little confusing regarding which group included overweight participants (i.e. BMI range of 25-30). From further reading, it appears overweight participants were allocated to the obese group, but it would be useful to the reader to specify this.

This section has been clarified accordingly. We added: “Therefore, when results were reported for obese, overweight and normal-weight individuals, obese and overweight individuals were combined in the obesity exposure group.”

3. "Contrary to what was initially planned, only studies in English or French were considered"... (Page 8). Can you explain what was originally planned and why a change was made?

We initially planned to apply no restriction on the language. However, during the study selection process, the limitation of human resources for translation forced us to revise the initial protocol regarding language inclusion. We recognize the potential publication bias associated with restricting language in systematic reviews; we thus found resources to translate the three studies that could not be translated earlier during the full text selection phase. None of these studies met the inclusion criteria. We corrected the sentences accordingly on p.8 and p.13. Figure 1 was also updated.

4. "Finally, 121 studies were included in the meta-analysis (20-22, 25, 51-152, 162-166, 168, 170-172, 174, 176-180): 13(153-161, 167, 169, 173, 175) were excluded because data was missing or could not be transformed"... I assume they were also excluded because data could not be obtained from authors? If so, this should be specified.

Indeed, five studies were excluded from the meta-analysis because data could not be obtained from the authors. This process is detailed at the bottom of page 9, but we added specifications in the results section (p.13-14). However, these studies were included in the descriptive analysis of the systematic review. 

5. You discuss T2DM and how that may have influenced your results in the discussion, but it may be worthwhile specifically mentioning cortical porosity given this appeared to be lower in those with obesity but has previously been shown to be increased in T2DM.

We thank the reviewer for this suggestion. We have added the increased cortical porosity in people with T2D in the introduction on p.5. We also added in the discussion on p.38 that cortical porosity has been shown to be increased in individuals with T2D and lower in those with obesity, which may explain, at least partly, the heterogeneity found for this outcome in people with obesity and the reported conflicting results. 

6. Paragraph 2 on page 35 essentially repeats the results and then itself is largely repeated on page 38; this could be removed or merged into the later section on BMD and bone microarchitecture.

We agree that results were repeated on p.35 and p.38. We removed parts of the results on p.38 to avoid repetition.

Reviewer #2

1. Although documents are provided somewhere else, it is better to briefly and clearly describe the protocol, especially search strategy, in the text for future readers.

We thank the reviewer for this suggestion. The protocol published in PROSPERO and the methods section of the manuscript contain the same information. We have now added some information in supplemental materials (i.e., describing the search strategy). To avoid repetition and be brief (in line with comment #2 below), keywords of the search strategy have been removed from the manuscript. 

2. It would be better to shorten the methods section, focus on objectives, and mention information in an easy-understanding order. For example, purpose and search strategy first, following by study selection, data extraction and quality assessment, and statistical analyses.

We recognize that the methods section is long. We reported our methodology and results according to the Cochrane review methodology and the PRISMA statement. The order and headings are also presented according to the PRISMA checklist. However, we changed the headings and order, as suggested, to mention the information in an order that is easier to understand.

3. In your protocol document registered with PROSPERO, there was no restriction on languages. Why did you exclude studies not in French or English in this study?

See the response to Reviewer 1’s comment #3.

4. The key metrics used in data analysis should be defined/explained and with formula if necessary. For instance: kappa static, funnel plot, inverse-variance, and quality of evidence. I2 statistic should be mentioned in bracket after kappa static at the first time for later use. GRADE approach needs to be briefly described, not only cited.

The kappa statistic is used to evaluate interrater reliability (study selection process), whereas I2 statistic is used to measure inconsistency of the effects between included studies or, in other words, test for heterogeneity (described on p.12). The reference for the kappa statistic is mentioned in the manuscript (ref. 37). To lighten the text, we decided to mention only the key metrics used in data analysis without the formula and add references for further details and explanations (kappa statistics and inverse-variance method). We agree that funnel plots for publication bias should be included for the readers, so we added figures in supplementary materials. We also briefly describe the GRADE approach on p.13.

5. Summary measures (Page 12): Choice of meta-analysis method is based on actual value of kappa static. It could be fixed-effects models (homogeneity, kappa ≤50%) or random-effects models (heterogeneity, kappa > 50%).

We appreciate this comment. However, we chose the random-effects method as we followed the Cochrane review methodology for data analysis recommendations. Consequently, the choice of meta-analysis method should be decided a priori and should be based on whether the exposition is expected to have truly identical effect or not. As stated in the Cochrane Handbook “Chapter 10: Analysing data and undertaking meta-analyses”, it is generally considered to be implausible that effects across studies are identical, unless the exposition has no effect at all, which leads many to advocate use of the random-effects meta-analysis. Moreover, the fixed-effect method ignores heterogeneity, which, in our case, would provide biased estimates due to the heterogeneity observed for most outcomes. Finally, the Cochrane Handbook states that “the choice between a fixed-effect and a random-effects meta-analysis should never be made on the basis of statistical test for heterogeneity [a posteriori]”. We also think the I2 should be used for assessing heterogeneity, rather than the kappa statistic.

6. Figure 1: Please include reason for excluding 8,914 records in the screening stage.

Reason for excluding 8,914 records in the screening stage has been added in Figure 1 (studies did not meet the eligibility criteria) based on PICOS. We acknowledge that it would have been interesting to add those details, but we followed the PRISMA methodology, which gives more importance on the reasons for full text exclusion only. 

7. Page 13: All articles included in or excluded from the research should not be cited in the text. They should be listed in a separate document as a supplementary file.

As reported in the Author Guidelines of PLOS ONE under the References section, we provided in the text the citation numbers associated with each reference in order that they appear in the text using the “Vancouver” style. As also stated in the PRISMA checklist (item 18), the citation should be provided for each included study. 

8. Page 26, 27: Please provide RR, 95% CI, p, and kappa, even there was no association between obesity and risk of fracture in studies combining men and women.

The results for the association between obesity and risk of fracture in studies combining men and women have been added. We added the RR, 95% CI, p and I2, since the kappa statistic does not apply for these results. 

9. Page 27: Several analyses showed low kappa (0%), it would be more appropriate to use fixed-effects model rather than random-effect models.

We considered this comment carefully but as explained earlier (see response to comment #5), we chose to follow the Cochrane Handbook for analysis methods.

10. Please provide funnel plot generated in the publication bias assessment to support your points in the result section.

We agree that funnel plots are of interest for the readers. We added the funnel plots in supplementary materials. 

11. Heterogeneity exploration: Please provide results for all outcomes, not only hip BMD.

We explored heterogeneity for all outcomes, but we did not find any other study to have a strong effect on the heterogeneity observed. We agree that it was not clearly stated in the heterogeneity exploration section, so we added a sentence to clarify this point on p.34. 

12. Discussion: Did authors consider the quality of the studies used in pool data? The low evidence of quality in the average results across the studies might be due to their different quality and confounding adjustments.

This is indeed a good point. Yes, we considered the quality of the studies in pooled data by conducting subgroup analyses based on the risk of bias of individual studies. The results of these subgroup analyses are reported in Supplementary Table 4. Moreover, the GRADE approach to assess quality of evidence includes the quality of studies (within-study risk of bias), directness of evidence, heterogeneity, precision of effect estimates and publication bias, which provides a complete evaluation of the quality of studies used in the pooled analysis.

---

## [Decision Letter · Decision Letter 1]

17 May 2021

Association Between Obesity and Risk of Fracture, Bone Mineral Density and Bone Quality in Adults: A systematic Review and Meta-analysis.

PONE-D-21-02000R1

Dear Dr. Gagnon,

We’re pleased to inform you that your manuscript has been judged scientifically suitable for publication and will be formally accepted for publication once it meets all outstanding technical requirements.

Kind regards,

Dr. Tuan Van Nguyen

Academic Editor

PLOS ONE

Additional Editor Comments (optional):

Thank you for your response to reviewers' comments and the revised manuscript. Both reviewers are happy with your response. I have no further comment. I consider that the manuscript is now suitable for publication.

Reviewers' comments:

Reviewer's Responses to Questions

**Comments to the Author**

1. If the authors have adequately addressed your comments raised in a previous round of review and you feel that this manuscript is now acceptable for publication, you may indicate that here to bypass the “Comments to the Author” section, enter your conflict of interest statement in the “Confidential to Editor” section, and submit your "Accept" recommendation.

Reviewer #1: All comments have been addressed

Reviewer #2: All comments have been addressed

2. Is the manuscript technically sound, and do the data support the conclusions?

Reviewer #1: Yes

Reviewer #2: Yes

3. Has the statistical analysis been performed appropriately and rigorously? 

Reviewer #1: Yes

Reviewer #2: Yes

4. Have the authors made all data underlying the findings in their manuscript fully available?

Reviewer #1: Yes

Reviewer #2: Yes

5. Is the manuscript presented in an intelligible fashion and written in standard English?

Reviewer #1: Yes

Reviewer #2: Yes

6. Review Comments to the Author

Reviewer #1: (No Response)

Reviewer #2: Thank you for well addressing all questions. I am satisfied with your response and totally agree that the manuscript is now good enough to publish.

7. PLOS authors have the option to publish the peer review history of their article (what does this mean?). If published, this will include your full peer review and any attached files.

Reviewer #1: No

Reviewer #2: No

---

## [Editor Report · Acceptance letter]

27 May 2021

PONE-D-21-02000R1 

Association Between Obesity and Risk of Fracture, Bone Mineral Density and Bone Quality in Adults: A systematic Review and Meta-analysis 

Dear Dr. Gagnon:

I'm pleased to inform you that your manuscript has been deemed suitable for publication in PLOS ONE. Congratulations! Your manuscript is now with our production department. 

Kind regards, 

on behalf of

Prof. Tuan Van Nguyen 

Academic Editor

PLOS ONE